# Comparative transcriptomics in serial organs uncovers early and pan-organ developmental changes associated with organ-specific morphological adaptation

Marie Sémon [1,3] ✉, Marion Mouginot [1], Manon Peltier[1], Claudine Corneloup[1], Philippe Veber [2], Laurent Guéguen[2] & Sophie Pantalacci [1,3] ✉

Mice have evolved a new dental plan with two additional cusps on the upper molar, while hamsters were retaining the ancestral plan. By comparing the dynamics of molar development with transcriptome time series, we found at least three early changes in mouse upper molar development. Together, they redirect spatio-temporal dynamics to ultimately form two additional cusps. The mouse lower molar has undergone much more limited phenotypic evolution. Nevertheless, its developmental trajectory evolved as much as that of the upper molar and co-evolved with it. Among the coevolving changes, some are clearly involved in the new upper molar phenotype. We found a similar level of coevolution in bat limbs. In conclusion, our study reveals how serial organ morphology has adapted through organ-specific developmental changes, as expected, but also through shared changes that have organ-specific effects on the final phenotype. This highlights the important role of developmental system drift in one organ to accommodate adaptation in another.

The evolution of new shapes arises from the evolution of their development. Current understanding of developmental evolution is still largely influenced by observations made by comparative developmental biologists in the 19th century and revisited in the 1980s by SJ Gould and others. They emphasized the parallels between development and evolution, caricatured by the formula "ontogeny recapitulates phylogeny", according to which successive evolutionary steps can be reflected in successive development steps, and the latest phylogenetic changes take root in the latest stages of development[1–3]. They also highlighted how changes in final shapes between species can be explained by so-called "heterochronies", that is, differences in the timing and duration of developmental

processes[3–5]. Today, we are still trying to characterize such patterns using modern tools such as transcriptomes to quantify developmental divergence. We also try to explain them using considerations about the nature of genes and gene regulatory networks that orchestrate development[3]. However we have made little progress in revealing the mechanistic principles of developmental evolution behind the old patterns[5], and more generally, behind the evolution of new shapes.

During development, shape emerges in a series of dynamic changes integrating different spatio-temporal scales (molecules, cells, forces in the tissue...). To move away from gene-centered views and understand shape formation as the product of a dynamic system, we

[1]Laboratoire de Biologie et Modelisation de la Cellule, Ecole Normale Superieure de Lyon, CNRS, UMR 5239, Inserm, U1293, Universite Claude Bernard Lyon 1, 46 allee d'Italie, F-69364 Lyon, France. [2]Laboratoire de Biometrie et Biologie Evolutive, Universite Claude Bernard Lyon 1, UMR CNRS 5558, 69622 Villeurbanne, France. [3]These authors contributed equally: Marie Sémon, Sophie Pantalacci. ✉e-mail: marie.semon@ens-lyon.fr; sophie.pantalacci@ens-lyon.fr

need to understand what rules this development trajectory and compare it between species. Indeed, what rules the developmental system may also create patterns such as heterochronies or the parallel between evolution and development.

The molar tooth is perfect for such a research program, because it combines a well-known evolutionary history, years of comparative developmental biology, mouse developmental genetics and a mechanistic understanding of the developmental system[6–8]. Here we take advantage of this model and a dedicated quantitative approach to characterize the developmental evolution of molars in mouse and hamster. We ask what changes in the developmental trajectory and in the developmental system produced a new molar shape in the mouse lineage. Below, we introduce the model system and the transcriptomic approach.

Molars develop from the physical and molecular interaction between an epithelium and a mesenchyme[8], Fig. 1a). The epithelium grows and folds to form the crown and its cusps under the influence of two types of signalling centres, PEK and SEK (Primary and Secondary Enamel Knots)[8]. First, the PEK determines the field of the molar crown. As this field grows, the SEKs are patterned sequentially and determine the cusps, starting with a buccal cusp[9–11]. This spatio-temporal sequence depends on activation-inhibition loops involving both epithelium and mesenchyme in a Turing-like mechanism operating while shape emerges[12,13]. Tooth morphogenesis models and in vivo experiments have shown that changes in the pathways controlling these loops can modify the number of cusps and recapitulate evolutionary changes[13–16].

Tooth evolution has been extremely well characterized in mammals thanks to an abundant fossil record. Mammalian molars generally have mountain-like shapes, with cusps, valleys and crests, which can vary in number and arrangement to perform different food processing tasks. Since the first mammals, lower and upper molars generally have different but interlocking tooth shapes[7,17,18]. The homologies of their cusps have been traced by paleontologists over hundreds of million years of mammalian evolution, sometimes with hot debates invoking developmental arguments[17]. Novelties in the number and arrangement of cusps have also been documented in many groups.

Interestingly, lab mice (and more generally the family of mice and rats) have a very specific upper molar morphology which is an evolutionary novelty. Between 18–12 million years ago (MYA), the upper molars of mouse ancestors gradually acquired two supplementary cusps on the lingual side, and reduced cusps size on the buccal side (Fig. 1b). This new morphology has been linked to changes in mastication and new dietary adaptations, facilitating the success of murine rodent radiation[19,20]. Mouse lineage stems from a cricetine-like ancestor, which lived about 25 MYA, and today's golden hamster is a good living representative of this ancestor's dental plan. Changes to the mouse lower molar dental plan were less drastic: cusp number was conserved, and changes were limited to reducing the lateral offset and changing connections between cusps to enable the new occlusion. To study this innovation, we therefore use hamster molars and lower molars as controls. We reasoned that lower molars would allow filtrating evolutionary changes specific to the upper molar phenotype, especially since some co-evolution of gene regulatory networks is expected between serial organs.

Developmental trajectories can be compared with transcriptomic time series. These have mainly proved useful to quantify the similarities of developmental trajectories between species (e.g. hourglass conservation pattern[21–25]. But more recently, we and others[10,26] have used transcriptomic timeseries to compare developmental trajectories in a highly integrated manner, pointing out heterochronies and possible morphogenetic mechanisms underlying an evolutionary novelty. For example, our previous work comparing mouse lower and upper molar development pointed out periods of maximal divergence and

heterochronies in the trajectories of the two teeth[10]. We showed how these transcriptomic variations correlated with variation in cell type proportions and were pointing to a high epithelium/mesenchyme ratio as a putative mechanism for upper molar specific shape[10].

Here, we compared the developmental dynamics of mouse and hamster molars, bridging transcriptomic divergence with divergence in developmental mechanisms. We found that the late formation of the two supplementary cusps is supported by early and complementary changes in development, impacting the whole dynamics. Our finding provides experimental validation to modern views on patterns of recapitulation.

We also made an unexpected finding since transcriptomes pointed to extensive developmental changes and co-evolution in the lower molar, including changes in developmental mechanisms that cause the supplementary cusps. We confirmed in another system, the mammalian limb, that extensive co-evolution of wing and limb transcriptomes accompanied the extreme phenotypic divergence of bat wing and legs, suggesting we uncovered a general principle of serial organ evolution.

## Results

### Supplementary cusps form last in mouse upper molar but this is only superficially a case of terminal addition

To compare the dynamics of cusp formation between molars and species, we first needed to account for the fact that mouse and hamster molars develop at different paces. We predicted developmental age from embryonic weight in each species and hybridized developing molars against a *Fgf4* probe to reveal PEK and SEKs (future cusp tips) from hundreds of samples. The patterns we observed among samples are consistent with a stereotypic sequence of stages corresponding to cusp patterning in each tooth and species (Fig. 1b, sides). Cusp patterning can be seen as a succession of irreversible stages representing step-wise cusp additions. The relative durations of these stages were estimated through continuous time Markov models as in ref.[10]. We then aligned temporal series between species with homologous start and end points of first lower molar morphogenesis (Fig. 1b, Supplementary Fig. 1).

Our results document the sequence of cusp patterning in mouse and hamster with an unprecedented level of detail, compared to previous studies in mice[15,27,28] and hamster[27,29]. As expected, the sequence of cups patterning is conserved in lower molars and is consistent with previous results[15,27]. The supplementary cusps of the upper molar form last, on the lingual side of the tooth, starting with the most posterior cusp (Fig. 1b). At first glance, the acquisition of the supplementary cusps seems to be a case of "terminal addition", where the development proceeds one step further in the organ with a new shape but earlier steps are conserved (orange arrows Fig. 1b). However, earlier changes are obvious in the spatial and temporal patterns of cusp addition in the upper molar, with a different 4-SEK pattern and a long 1-SEK stage (blue and grey arrows). A focus on the four posterior cusps, that are strictly homologous, confirmed these early changes (Supplementary Fig. 2). Therefore this is only superficially a case of terminal addition.

### Gene expression dynamics are already divergent in the early steps of morphogenesis

To quantify temporal variation in molar morphogenesis, we obtained RNA-seq data at high time resolution. Data was sampled and processed to be as comparable as possible between species (Fig. 2a). Yet, in a PCA analysis that separates samples according to the main axes of variation in the data, the first principal component is associated with mouse/hamster difference (47.8% variance), followed by development time (10.2%). Coordinates on this time axis confirmed the homology of samples chosen for homologizing the time series in this study. Upper and lower molar samples are only separated from each other on the sixth component, which carries 3% of the total variance. This PCA does

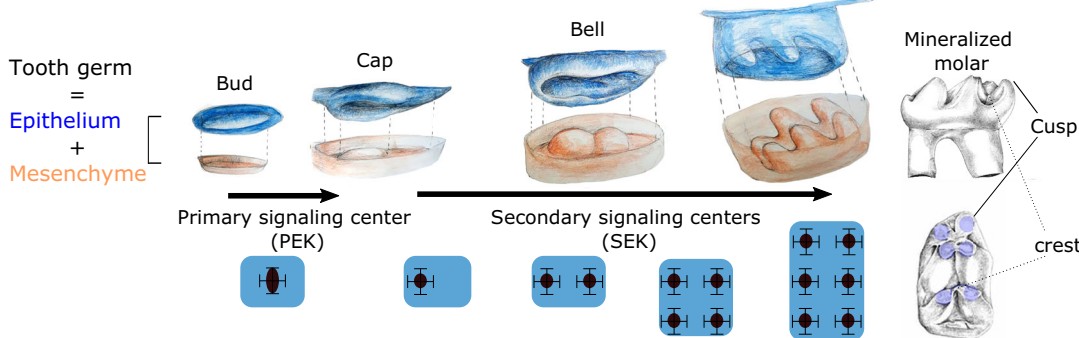

## a. Morphogenesis

## b. Cusp patterning

**Fig. 1 | The evolution of the new cusps in the mouse upper molar involved changes in the whole dynamics of cusp patterning. a** Drawings of the epithelium and mesenchyme compartments at four stages of molar development: at "bud" stage, an epithelial signalling centre called PEK (Primary Enamel Knot) triggers the formation of a "cap" defining the future crown. Between "cap" and "crest" stage, SEKs (Secondary Enamel Knots) are patterned sequentially in the epithelium and drive cusp formation. By the end of morphogenesis, the mesenchyme has the shape of the future tooth and the epithelium is a dental impression. **b** Final shapes, dynamic and pattern of cusp addition in mouse and hamster lower and upper molars. Sides: Upper views of final morphologies represented by 3D scans. Supplementary cusps in mouse upper molars highlighted in orange. Panels: Series of developing molars (dots) hybridized against *Fgf4* to reveal signalling centres. Time series were modelled using Markov processes as a series of stages with specific durations (blue bars). x-axis: developmental age, y-axis: morphological stages, with schematized SEK arrangements. Dotted lines: homology of early cap and crest stage. Orange, blue and grey arrows show 3 differences in hamster and mouse upper molars.

not show an excess of variation at the end of the sequence of cusp formation, as expected for classical terminal addition (Fig. 2b).

To directly quantify differences along morphogenesis, we modeled temporal expression profiles in each molar with polynomials (Fig. 2c). For each gene we fitted four distinct curves, one per tooth. We measured the distance between pairs of curves in ten time windows over development. Among all possible pairs, the distance between upper molar transcriptomes is highest among all pairs of molars, as

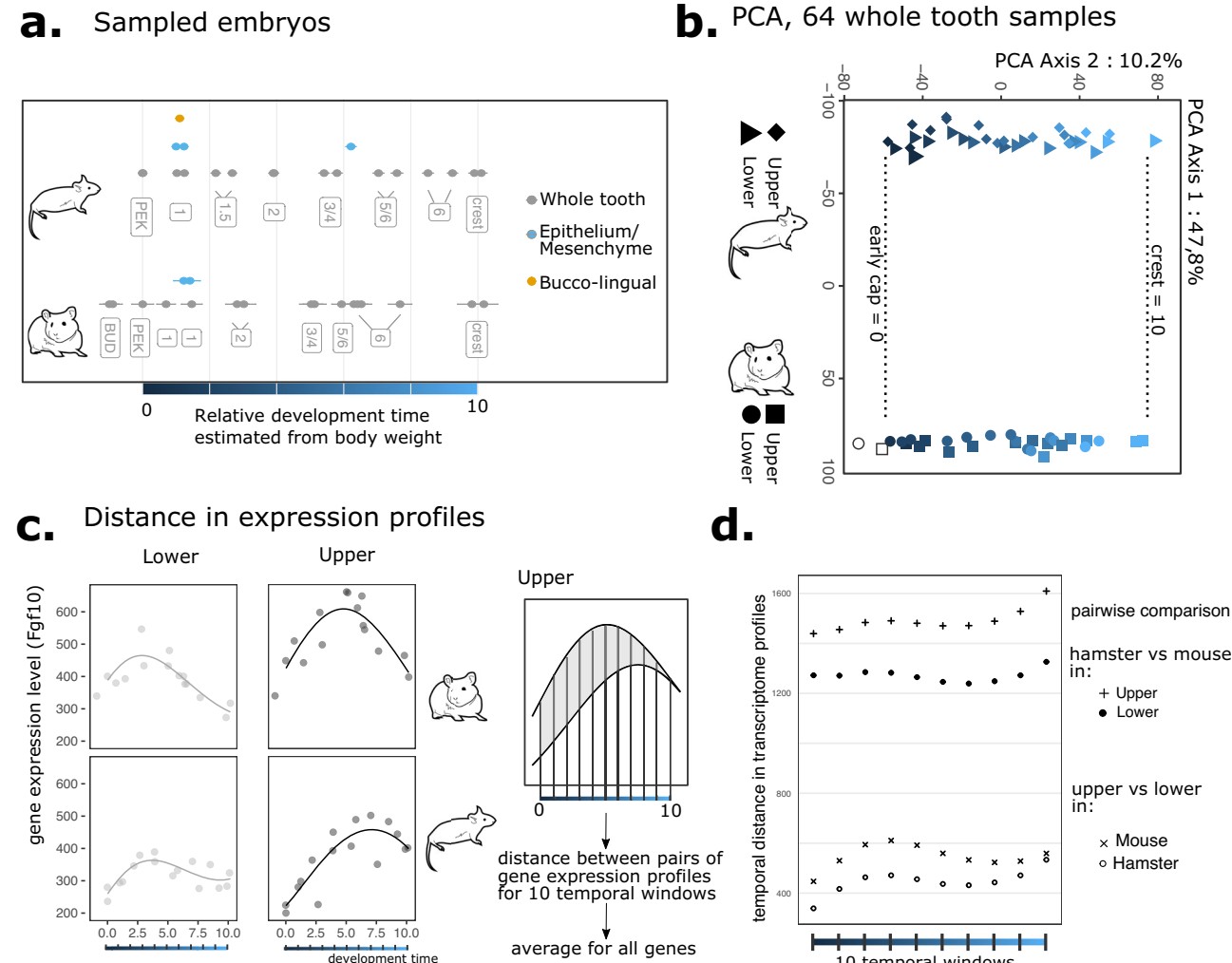

**Fig. 2 | The dynamics of gene expression has changed throughout the whole tooth morphogenesis in mouse and hamster. a** Embryo sampling used for expression profiling, in whole tooth germs, tooth tissues (mesenchyme/epithelium) and tooth halves (bucco/lingual). Each embryo provided upper and lower samples. Relative developmental time established from embryonic weight and boundaries for stage homology as in Fig. 1. The morphological stages (as in Fig. 1) indicated are for lower molars. **b** Principal component analysis (PCA) of 64 whole molar bulk transcriptomes based on 14532 1:1 orthologues. Each symbol is an individual transcriptome with a colour gradient for the embryo's relative development times. Dotted lines: Stage homology established by morphology is confirmed by transcriptomes and used as boundaries for relative developmental time. **c** Temporal expression profiles modelled with polynomials for the *Fgf10* gene. Distance between pairs of curves is computed for ten time windows (comparison of hamster and mouse upper molars profiles is shown for *Fgf10*). **d** Average distance in each window shown for all pairwise comparisons.

expected from the difference of morphologies. It remains rather constant during development, but shows a slight increase at the first third and toward the end of our dataset (Fig. 2d). Divergence in upper molar transcriptomes, like divergence in spatial and temporal patterns of cusp addition, is therefore only partially in accordance with terminal addition. Hence, the whole transcriptome trajectory has evolved, particularly in early stages.

**Developmental gene expression shows that several aspects are modified in the early morphogenesis of the mouse upper molar**

We used the transcriptome time series as a starting point to investigate several changes of development specific to the mouse upper molar that may favor its additional lingual cusps.

Our previous results showed the mesenchyme:epithelium ratio is increased in the mouse upper molar as compared to the lower molar[10]. This may favor additional cusps, because in artificial teeth made by reassociating mesenchymal cells to a single epithelium, the number of cusps formed increases with the number of mesenchymal cells[30]. This hypothesis only holds if the mesenchyme:epithelium ratio is similar in upper and lower molars in hamster. To check this, we extracted all mesenchyme and epithelium-specific marker genes from tissue-specific transcriptomes (Fig. 2a), and used in silico deconvolution to estimate the mesenchyme proportions from whole tooth germ transcriptomes (Fig. 3a). The mesenchyme:epithelium ratio was indeed significantly higher in the upper molar in mouse, but not in hamster (Wilcoxon tests, $p < 2e\text{-}16$ and $p = 0.152$), which controls that the increased proportion of mesenchyme is specific to the mouse upper molar. This was confirmed by direct quantification of mesenchyme:epithelium ratio performed on 3D reconstructed tooth germ at an early stage (Fig. 3a and Supplementary Table 2).

As seen above, the divergence of the upper molar transcriptomes peaked in the first third of morphogenesis (Fig. 2d). This corresponds to the end of the 1-SEK stage, which is longer in the mouse upper molar than in any other tooth (likelihood ratio test, $p < 1e\text{-}16$, Fig. 1b; see also later Fig. 4b). Intriguingly, at that stage, the tooth germs grow rapidly on their lingual side, where supplementary cusps will form later.

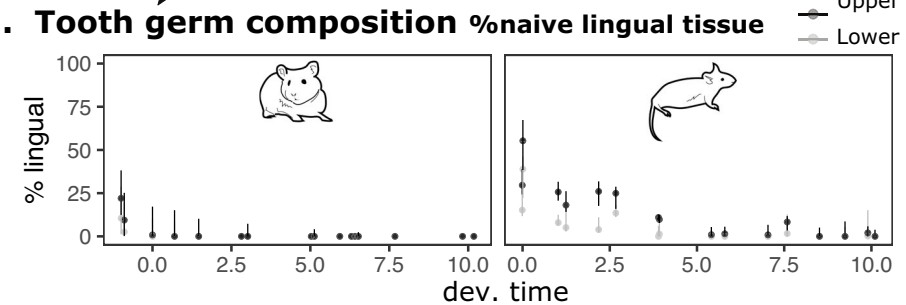

**a. Tooth germ composition** mesenchyme *vs* epithelium

**b. Lingual growth duration**

**c. Levels of pathway activation in buccal/lingual part**

**d. Tooth germ composition** %naive lingual tissue

**Fig. 3 | Several early changes, including heterochronies, correlate with supplementary cusps in the mouse upper molar. a** Percentage of mesenchymal tissue in tooth germs estimated by deconvolution of the RNAseq time series with mesenchyme and epithelium-specific marker genes (left) and with direct volume quantification in 3D reconstructed tooth germs (right). Time scale in relative 0–10 scale (x axis) for each species. Color code indicated for each molar. **b** Relative duration of the first morphological stages highlights a longer period of lingual growth in mouse upper molar (1-SEK, purple). Stage durations from Fig. 1b. PEK and SEK: primary and secondary enamel knots respectively. **c** Levels of activation of BMP, SHH and WNT pathways in buccal and lingual sides of the mouse molars at 15.0 dpc (1-SEK stage). Measurements made with an in silico method, ROMA, using two lists of target genes to estimate both an epithelial (epi) and a mesenchymal (mes) pathway activity in the 15.0 buccal and lingual RNAseq samples. Drawing on the left represents the dataset design. **d** Proportion of naive lingual tissue in mouse and hamster molars estimated by deconvolution of the RNAseq time series with lingual and buccal marker genes.

This finding prompted us to look into specificities of the bucco-lingual development in the mouse upper molar. We obtained mouse transcriptomes of buccal and lingual halves at the early 1-SEK stage and estimated how strongly the three pathways controlling cusp formation were activated in these samples (WNT, BMP4 and SHH) with ROMA[31]. All three pathways were strongly activated on the buccal halves, which start to pattern cusps earlier. They were activated at intermediate levels on the lingual half of the lower molar, but the upper molar lingual half remained naive (Fig. 3c). Thus, both early tooth germs are polarized, but polarization is stronger in the upper molar. *Osr2* and *Sfrp2* are two genes known to limit the formation of additional lingual teeth in the mouse jaw by inhibiting the Wnt pathway[32–35]. They showed polarized expression in the lingual part of the tooth germ (Supplementary Fig. 3). Their degree of polarization and timing of downregulation correlated with

formation of lingual cusps (first in lower, then in upper molar). This suggested to us that the early polarization of tooth germs by Wnt inhibitors transiently prevents tissue induction on the lingual side, which protects it from associated growth arrest, and enables lingual growth to form a second lingual cusp. Increased polarization may result in an heterochrony of this mechanism in the upper molar, favoring supplementary cusp formation specifically on the lingual side.

To test this idea, we quantified the proportion of naive lingual tissues in mouse and hamster tooth germs, by deconvoluting the time series dataset with buccal and lingual tissue marker genes[36]. As expected given the progressive nature of cusp formation, the proportion of naive lingual tissue decreases during morphogenesis in both species (Fig. 3d). But in mouse molars, and even more markedly in the mouse upper molar, the initial proportion of naive tissue is larger,

## Upper/Lower expression profiles co-evolution

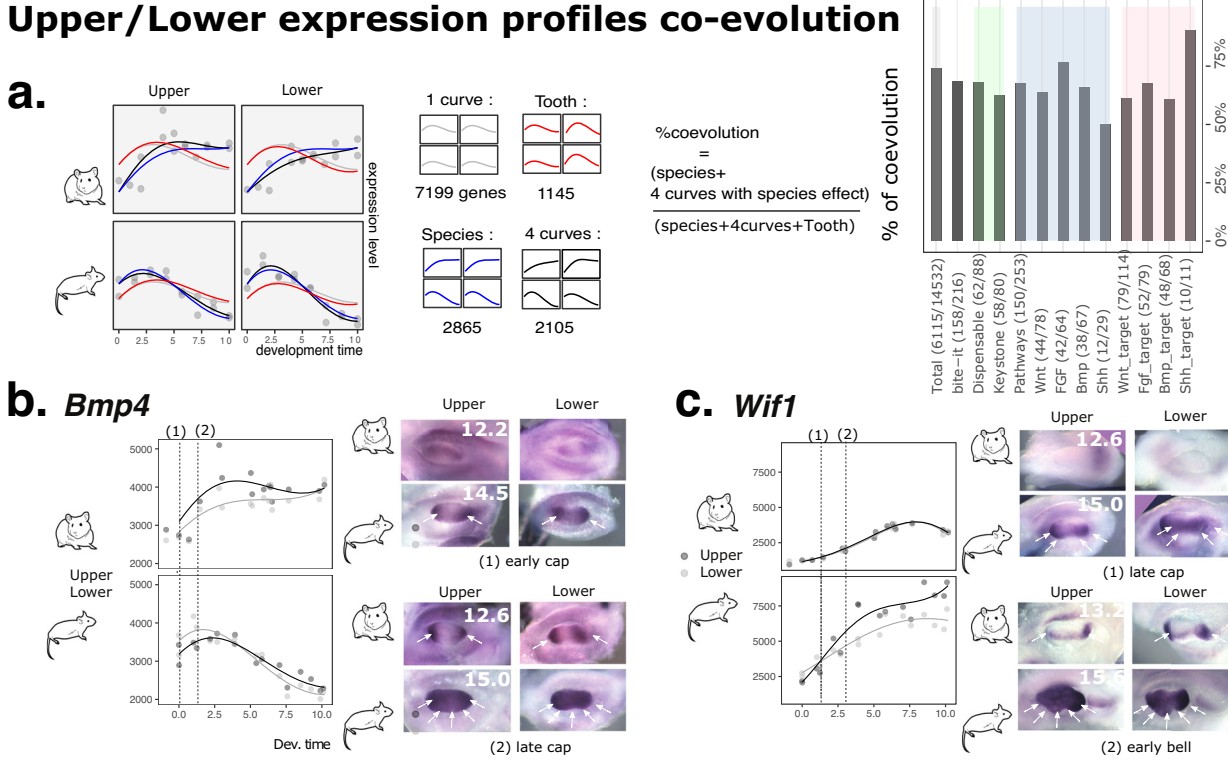

## Hamster-mouse expression profiles divergence

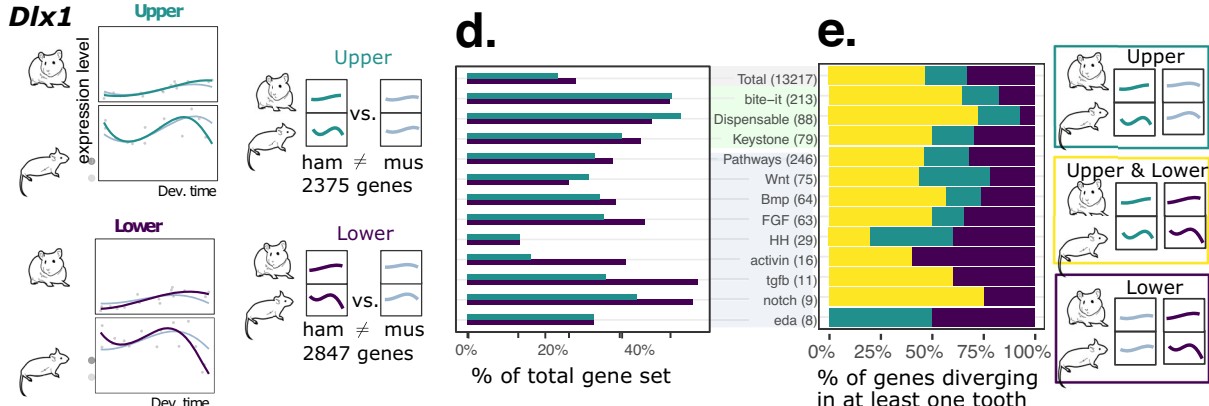

**Fig. 4 | Expression profiles have coevolved extensively and diverged as much in lower as in upper molars. a** Nested models of temporal profiles taking the four teeth altogether. The percentage of coevolution is computed as a proportion of informative genes varying between species and/or teeth, for different gene categories with numbers into brackets (number of informative genes/total number of genes in the category). Dispensable and keystone gene categories taken from ref.[37]. Genes from tooth developmental "pathways" (further splitted into individual pathways, blue) and their target genes (red) taken from refs. [96]. **b, c** *Bmp4* and *Wif1*'s expression profiles corroborated by in situ hybridization of dental mesenchyme show that expression in upper and lower molar has coevolved. Dashed lines map pictures to the timeseries and numbers in the picture corners are

developmental age. Arrows point to regions of the dental mesenchyme with strong expression. See supplementary Fig. 5 for details. **d** (grey dots), and models (curves). Top: the "upper divergent" model, allowing different profiles in mouse and hamster (green), is compared with the "upper non-divergent" fitting the same profile but different baseline expression levels (grey). Bottom: Same models fitted independently for lower molars (purple and grey). Best model was chosen for each molar by likelihood ratio test (adjusted $p < 0.05$). Barplots: percentage of divergent profiles in upper and in lower molars (categories as in a). **e.** Percentage of the "divergent" genes detected in d, found both in upper and in lower molars (yellow), only in upper (green) or only in lower (purple).

and diminishes more slowly (in accordance with *Sfrp2* expression, Supplementary Fig. 3).

Hence, we found two changes mirroring the divergence of upper molars transcriptomes: a change in mesenchyme proportion that is already present early and run throughout development and an heterochrony in early development (naive tissue maintenance) likely stemming in an early change in bucco-lingual polarity.

### Gene expression dynamics largely co-evolved between upper and lower molars

In the PCA analysis, the main axis of variation in the transcriptomes separates the samples by species, but groups upper and lower molars (Fig. 2b) which hints that gene expression co-evolves between molars. This may be caused by differences of basal expression levels or this may reflect co-evolution of the temporal dynamics of morphogenesis.

To confirm and quantify the later possibility, we with fitted four models to the temporal profiles (Fig. 4a, Supplementary Data 1): The most complex model has four curves (one distinct per tooth, as in Fig. 2d), intermediate models have two curves (distinguishing species: hamster/mouse or distinguishing tooth: upper/lower), and the most simple model has a single curve common to all teeth (1 curve). Models for different species account for different baseline expression levels, to make sure that we focus on species differences in temporal dynamics. We attributed the best model to each gene. 6115 genes (42% of our dataset) were informative, showing some differential temporal regulation between tooth and/or species (i.e. the 1 curve model is outcompeted by another, more complex model). We then built an index of coevolution, as a percentage of these informative genes showing co-evolving profiles (Fig. 4a and methods). We estimated that the expression profiles of 74% of these informative genes have coevolved (34% of the dataset).

We found that several processes are overrepresented in these co-evolving genes, some of which may represent species-specific heterochronies in cell differentiation or colonisation (eg by nerves and blood vessels), or size regulation (IGF1 pathway), but others are clearly associated with morphogenesis (cell and tissue migration) (Supplementary Fig. 4). This index is also high among genes important for tooth development (e.g. see keystone genes[37] in Fig. 4a, which all have a strongly detrimental effect on tooth formation). The main tooth signaling pathways, and their targets, are concerned, suggesting that the whole dynamics of tooth morphogenesis has co-evolved.

This coevolution came as a surprise but was confirmed for genes of key pathways by in situ hybridization. The expression of the keystone gene *Bmp4* peaks earlier in mouse transcriptomes than in hamster: its mesenchymal expression also reaches earlier a spatially homogeneous expression in the mouse tooth germs (Fig. 4a, Supplementary Fig. 4). The expression of *Wif1*, a known modulator of the Wnt pathway critical for cusp formation, rises earlier in mouse transcriptomes, and its mesenchymal expression rises earlier and invades a larger territory in mouse tooth germs (Fig. 4c, Supplementary Fig. 4). The expression of *Dkk1*, another Wnt inhibitor, restricts and focuses earlier in both mouse molars, at the future cusp tips (Supplementary Fig. 5).

## The co-evolution of transcriptomes reflects co-evolution of morphogenesis, as shown by the co-evolution of signaling centers dynamics and spatiality

Because the expression of many genes has co-evolved, including those of genes that control cusp formation, we hypothesized the dynamics of cusp patterning may have coevolved as well. Indeed we quantified that both mouse molars quickly transition to 1-SEK after a rather short PEK stage (Fig. 1, Fig. 5a). By comparing epitheliums of the two species matched for growth advancement, we confirmed that mouse molars switch earlier to 1-SEK: Both mouse molars already exhibit the rounded and focalized *Fgf4* expression typical of a SEK when hamster's still exhibit the large and elongated *Fgf4* expression typical of the PEK (Fig. 5c stage 2). The dynamics of signaling centers patterning thus evolved in a concerted manner in mouse molars, with anticipated cusp formation in both mouse molars.

Spatial aspects of cusp patterning also show concerted evolution. This can be seen from the expression of 3 diffusing signals, which are produced in the SEK and inhibit the formation of other SEKs in the vicinity. We found that *Bmp4*, which acts in the epithelium as a cusp formation inhibitor ([38]), is expressed from the signalling centers with a more narrow and roundish pattern in mouse than in hamster (both in PEK and SEKs, Fig. 5b, Supplementary Fig. 6). Two other inhibitors, *Shh*[39] and *Wif1*[40,41] also show a more restricted expression in mouse (Supplementary Fig. 6). Hence, inhibition is more local in both mouse molars.

In conclusion, both dynamics and spatiality of activation-inhibition mechanisms have evolved in a concerted manner in the molars of the two species. Both of them, rapid switch to SEK formation relative to epithelial growth and more local inhibition, are predicted to favor the formation of more cusps. These developmental features, that make sense for the formation of the supplementary cusps in the upper molar, are thus surprisingly observed also in the lower molar. We therefore reconsidered findings from Fig. 2, and realized that at least two other features consistent with supplementary cusp formation, the long 1-SEK stage and late maintenance of naive tissue, have also co-evolved in the lower molar, although the developmental phenotype is milder.

## Lower molar trajectories of developmental gene expression evolved as much as upper molar ones, in a case of developmental system drift

Through transcriptomes and marker gene expression, we uncovered several developmental phenotypes of the lower molar that have evolved in concert with the upper molar. Yet, this evolution did not drive a major phenotypic change. Such discrepancy between the divergence of development and the conservation of the final phenotype, is a phenomenon known as Developmental System Drift (DSD). To measure the extent of this phenomenon, we decided to compare levels of developmental evolution in both teeth. Since the lower molar phenotype has been much more conserved during evolution, the lower molar developmental phenotype captured by the temporal profiles should be more conserved. Otherwise, this is an indication of DSD.

We scored the divergence between mouse and hamster upper and lower molars by modelling temporal profiles with polynomials (LRT with adjusted $p < 0.05$). We found that for 21.4% of genes, the profiles have diverged in the lower molar, which is even more than in the upper molar (17.5%, $p < 10^{-9}$ for a two-sided proportion test, Fig. 4d, Supplementary Data 1). This is true as well for genes relevant for tooth-development and phenotype ("bite-it", "keystone", "pathways"; Fig. 4e).

Put together, these observations suggest that the development of the lower molar has drifted while co-evolving with that of the upper molar.

## Similar co-evolution is observed in bat limbs transcriptomes

In order to test the generality of our findings beyond mouse molars, we turned to bat limbs, another pair of serial organs with drastic changes in one appendage but not the other. The evolution of the wing involved many changes in the forelimb development, including digit patterning, growth, and webbing to form the wing membrane. In comparison, the bat hindlimb kept a morphology more typical of quadrupedal species, as did both mouse limbs, and both have been taken as controls in previous studies[42].

We collected raw sequencing data from a previous study comprising 3 stages of mouse and bat fore/hindlimb development (Maier et al. 2017) (Fig. 6a). We quantified expression levels and classified temporal profiles with polynomial models dedicated to measure coevolution (as in Fig. 4a). The profiles of 714 genes differed both between species and limbs. The profiles of almost four times more genes (2677) diverged between the two species, but co-evolved in the two limbs, despite their drastic morphological differences. Such a large proportion of co-evolving genes mirrors our finding in rodent molars. Importantly, the temporal dynamics of genes with a well-established role in controlling limb morphology have co-evolved. It is the case of key genes controlling limb patterning (*Shh*, *Fgf10*, *Fgf8*, *Grem1*…) and chondrogenesis (*Wnt3* and the Activin pathway: *Inhba*, *Inhbb*, *Acur2b*…). It is also the case of most of the genes known to regulate webbing (*Fgf8*, *Grem1*, *Bmp7*, *Ihh*, Retinoic acid pathway: *Aldh1a2*, *Cyp26b1*).

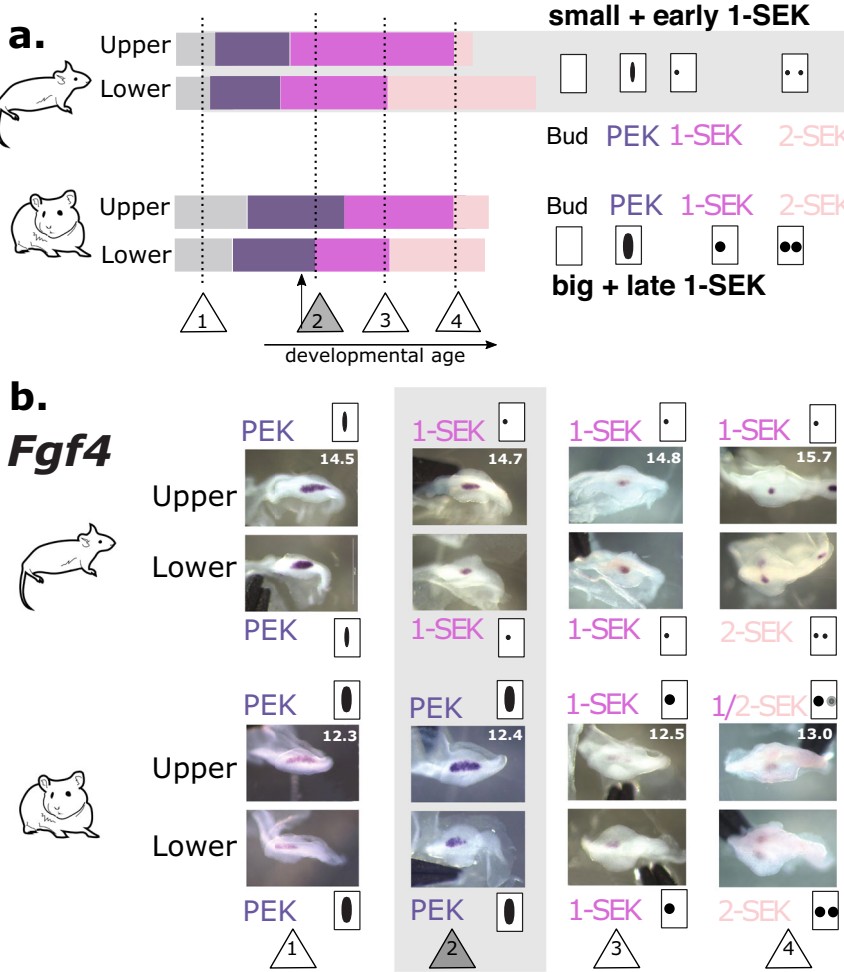

## Small-range signalling centers in mouse

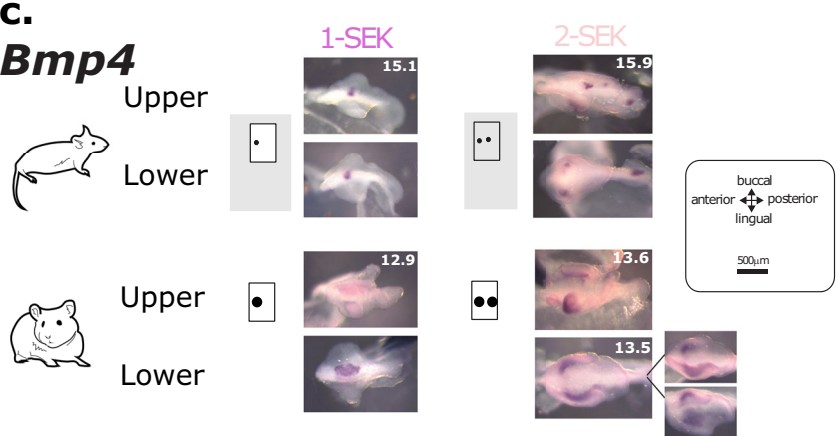

**Fig. 5 | The spatio-temporal dynamics of activation-inhibition mechanisms has co-evolved in mouse molars. a** Duration of each stage estimated by Markov models from Fig. 1b. Numbers in triangles as in **b**. Timeline and cartoons on the right recapitulate changes in signalling centres. **b** Expression of *Bmp4* is more focalized in mouse than in hamster SEKs. Mouse and hamster samples are paired for similar advancement of epithelial growth. Age of samples in days is in the upper molar picture (for samples taken from the same embryo) or in both pictures

(samples taken from different embryos). See also Supplementary Fig. 6. **c** Mouse molars transition earlier to cusp patterning. Transition from the PEK ( = 0 in the matched timeline) to the 2-SEK stage as seen on tooth germ epithelial parts hybridised against *Fgf4*. Pairs of mouse/hamster embryos were selected to show four remarkable steps in this chronology (1–4 in triangles). At stage 2, *Fgf4* expression is still elongated in hamster, as typical for a PEK, while it is already roundish in mouse, as typical for a SEK.

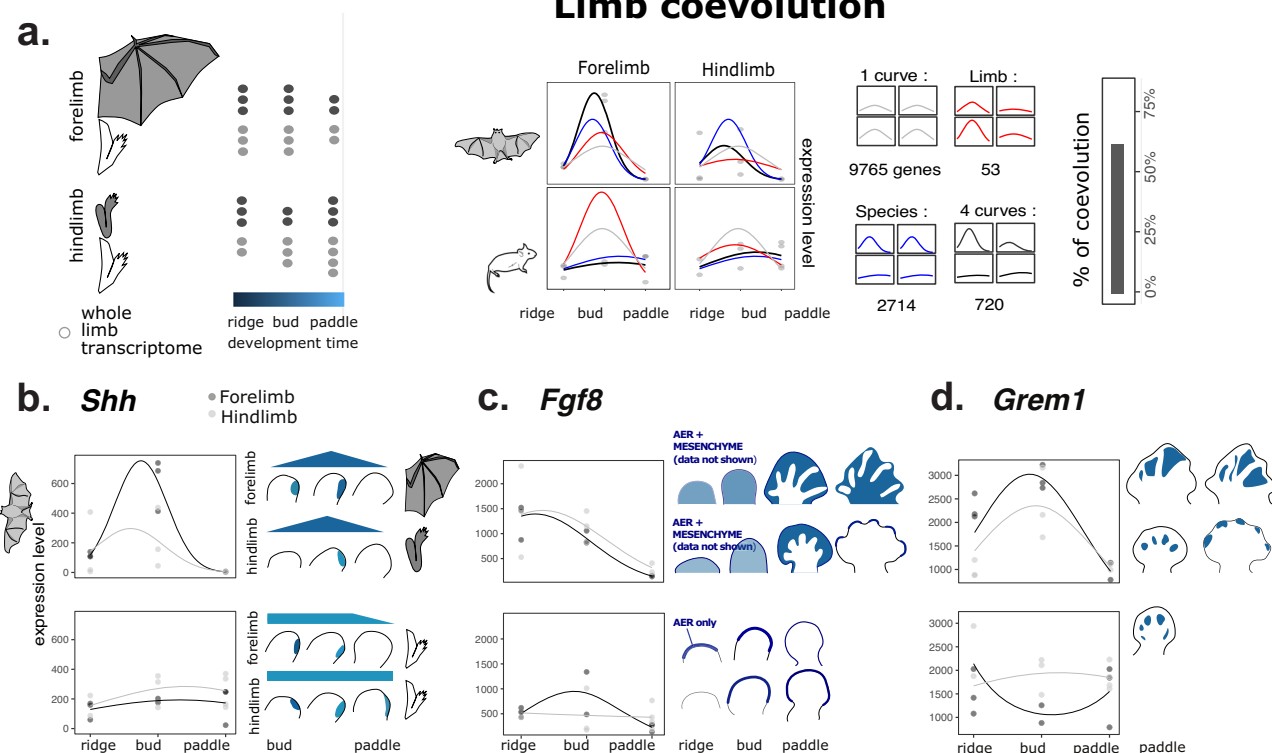

**Fig. 6 | Expression profiles have coevolved extensively in bat limbs, including for genes involved in wing evolution. a** Transcriptome dataset reused from (Maier et al. 2017) with fore- and hindlimb samples at the ridge, bud and paddle stage for the bat *Carollia perspicillata* (stages CS13, CS14, CS15) and the mouse (E10, E10.5, E11.5). Coevolution of fore- and hind limb expression profiles was measured with nested models as in Fig. 4. *Shh* gene models are shown together with global quantifications. **b** *Shh* expression profiles from transcriptomes and published in situ hybridizations patterns, redrawn from ref.[43]. Bat species is *Miniopterus natalensis*, mouse stages are E11.0, E11.5, E12.0. In both transcriptome and in situ hybridization, *Shh* expression is peaking in bat limbs as compared to mouse limbs. **c** *Fgf8* expression profiles from transcriptomes and published in situ hybridizations patterns. Mouse patterns drawn from ref. [101] and[102], and bat patterns from ref. [44]. In bats, a new domain of *Fgf8* expression is observed in the mesenchyme from early stages, on top of the conserved expression of *Fgf8* in the apical ectodermal ridge (AER) of both mouse and bats. In the latest stage, this mesenchymal expression is maintained in the wing only. **d** *Gremlin1* expression profiles from transcriptomes and published in situ hybridizations patterns. Mouse patterns drawn from ref. [101] and bat patterns from ref.[44]. *Grem1*, as *Fgf8*, co-evolves in early limb transcriptomic dataset. However, in later stages, published patterns show a wing-specific expression, while the bat leg has kept a similar expression profile as in mouse anterior limb.

We note that several of these genes have been pointed in the literature as key for bat wing evolution. For three of them, we could compare the expression profile in the transcriptomic dataset with published in situ hybridization in both limbs ([43],[44]; summarized in Fig. 6b–d), and they were consistent with co-evolution. The iconic *Shh* gene expression clearly peaks at the second stage in both bat limbs, but not in mouse limbs (Fig. 6b), and peaking is exaggerated in the bat forelimb. The new temporal profiles of *Fgf8* and *Grem1* in both bat limbs are also consistent with a previous study, which has reported the novel expression domain of these genes in the mesenchyme of both limbs at these stages and/or slightly later during webbing (Fig. 6c, d).

As in mouse molars, co-evolution is pervasive in bat limbs and some developmental features which were thought to be key for the new morphology are also concerned.

## Discussion

Here we aimed at an integrated understanding of morphogenesis evolution in a case of two serial organs, where one changed shape but not the other. By inspecting the general patterns of transcriptome conservation and by analyzing the underlying developmental mechanisms, we rule out two common expectations on the evolution of development. First, we reveal unsuspected early heterochronies, and start bridging them with cellular and molecular processes. Second, more strikingly, we revealed the marked developmental co-evolution of the lower molar, which we had first introduced as a control. Far from

that, it shares with the upper molar several developmental phenotypes that are predicted to favor supplementary cusp formation. This shows that the developmental co-evolution of two organs is not antithetic to their independent morphological evolution, and can even be directly involved in the independent evolution of a morphological novelty. Below, we discuss our three main findings, and finally, the implications for our conceptualization of developmental system drift, concerted evolution and pleiotropy.

First, our work indicates that the new mouse phenotype likely involved multiple complementary changes of the patterning system. Theoretical studies propose that the patterning of teeth and cusps are determined dynamically at two levels: the parameters of a Turing system (activator, inhibitor and their interaction) and the field size (control of growth and/or competence)[45–47]. In tooth and other systems, many examples of evolutionary changes targeting one[13,15,48] or the other[49–51] have been suggested. Here we propose that the mouse upper molar innovation relied on three complementary developmental changes at these two levels (Fig. 7).

1- Epithelial inhibitors are produced more locally which helps squeezing more SEKs in an equivalent field. *Bmp4* is of special interest since increasing *Bmp4* levels in the mouse epithelium suppresses the supplementary cusps[38].

2- The bucco-lingual polarization is increased with a larger naive field on the lingual side, possibly due to increased Wnt inhibition. This large naive tissue could be responsible for the rapid and sustained

**Fig. 7 | A summary of findings and working model from this study for mouse molars and bat limbs. a** Two pairs of serial organs where one organ underwent drastic shape changes as compared to the relative conservation of the other. **b** Developmental stages covered by transcriptome data with key changes associated with supplementary cusps (molars) and webbing (limbs). Transcriptomes are dominated by co-evolution. Such shared changes (e.g. smaller signalling centers in molars) are necessary but not sufficient. In combination with specific changes (e.g. mesenchyme size) they allow to reach a threshold for organ-specific adaptation. In mouse molars, 3 changes (increased mesenchyme size, increased bucco-lingual asymmetry and smaller inhibitory signalling centers) combine to induce extra cusps on the lingual side of the upper molar only. In bat limbs, *Fgf8* and the BMP inhibitor *Grem1* combine to suppress interdigital apoptosis in the forelimb only[44]. In mouse, a combined increase of FGF signaling and decrease of BMP signaling is necessary to suppress apoptosis and induce webbing. **c** Patterns of developmental divergence. top: Upper molar supplementary cusps develop last. We could expect divergence to accumulate during development. Lower molar divergence was expected to be much smaller. bottom: The observed levels of upper and lower molar divergence are comparable and the onset of the three morphogenetic changes described in b happen early in development.

epithelial growth of the lingual side providing a larger field to fit SEKs over the course of morphogenesis.

3- The mesenchyme:epithelium ratio is increased, which is known to promote cusp formation[30]. This may be because the mesenchyme stimulates epithelial growth and/or more directly produces an activator of EK formation[52]. However two obvious candidates, *Activin βA* and *Fgf3*[53,54], showed no expression change (Supplementary Fig. 7).

We believe that these changes act in synergy to reach the full phenotype for several reasons. 2- (but not 1- and 3-) can specifically increase cusp number on the lingual side. The difference in ratio observed in 3- seems far from what would be needed to add two more big cusps, when compared to ratios used in synthetic teeth[30]. Finally, supplementary cusps in mouse mutants are typically limited to small accessory cusps, and even though increasing cusp number is possible in vitro by treating cultured teeth, it is better achieved by playing on multiple molecular inputs[16]. The successive acquisition of the three changes during evolution may correspond to the steps seen in the fossil record, from a very small to a larger lingual cusp, to two supplementary cusps with reduced buccal cusps.

Second, our work highlight how the new ontogeny of mouse upper molar recapitulates phylogeny despite involving early changes, with several implications on the relationship between ontogeny and phylogeny. The two new cusps form late and last in development, they even pattern one after the other in an order that recapitulates their appearance in the fossil record[19,20]. Hence, at first glance, the evolution of mouse supplementary cusps follows the expectations of a scenario of recapitulation by terminal addition, as defined by Gould 1977[2]: the latest steps of ontogeny are implemented during evolution to produce novelty, and as a consequence, ontogeny recapitulates phylogeny. Our integrated study of the whole developmental sequence however points to a very different hidden scenario (Fig. 7c). We found that transcriptome divergence is high and nearly constant throughout development, with a small peak at the first third of morphogenesis and a small increase at late stages. This matches the temporal distribution

of the presumptive causative changes in the mouse upper molar. The mesenchyme:epithelium ratio is constantly biased, and two heterochronies manifest in the first third of morphogenesis: earlier onset of transition to 1-SEK stage (pre-displacement), and late offset of B/L polarization with lingual naive tissue (hypermorphosis). Settings of activation-inhibition differ in SEKs from the very first patterned SEK. As a consequence, the spatio-temporal acquisition of EK and SEK clearly differed, most notably in the PEK, 1-SEK and 4-SEK stages. Thus, mouse upper molar evolution is only superficially a case of recapitulation by terminal addition. Instead, changes in the settings of the system influence the whole trajectory of EK formation from earliest stages to support the late addition of two cusps.

Our findings therefore provide an empirical support to theoretical predictions on the evolution of tooth morphogenesis, and dynamical systems in general. Indeed, the morphodynamic model of tooth morphogenesis predicts that differences in activation-inhibition parameters and/or growth parameters will dynamically interact through cascade effects to produce additional steps of cusp formation[7,55]. This model has been abundantly used to predict evolutionary changes and to recapitulate natural patterns, including the occurrence of supplementary cusps on the border of the tooth (among others[13,15,48,55–57]. However, to our knowledge, it is the first time that its foundations are validated by the experimental comparison of tooth formation in two species.

In the Haekelian view of recapitulation, the parallelism between development and evolution occurs because terminal changes in development are added during evolution. Recent theoretical work proposed that recapitulation rather occurs because evolution has selected a developmental system whose variation properties were built by its evolutionary history[58]. This is exactly how we interpret the evolution of the supplementary cusp in mouse. Molar morphology has complexified during early mammal evolution, progressively adding cusps to more simple teeth, and this built the logic of the molar developmental system found in true mammals. The parallelism seen in

mouse development and evolution for the two supplementary cusps is a product of the logic of the developmental system[7].

The above theoretical work also makes predictions on heterochronic changes, which could be tested in our setting. Indeed, our study also suggests how multiple patterns of heterochronies in the morphogenetic process may in fact root in the control of early organ size, polarity and in the regulatory feedback whereby the tooth self-organizes. This will shed light on the molecular and cellular nature of heterochronies which remains unclear[5].

Third, we show how developmental and gene expression changes associated with a new phenotype in one organ are partially shared with others. Since we used the lower molar as a control, we had two obvious expectations. First, that the development of the organ with a new shape, the upper molar, evolves more than the development of the control organ, the lower molar. Second, that developmental innovations causing the new phenotype are specific to the upper molar. However, we show that the lower molar development has evolved as much as, and coevolved with, the upper molar (Fig. 1d and Fig. 4d). Part of this may be associated with coevolving morphological features like cusp height and slope, absence of a longitudinal crest linking cusps in mouse. Another part may be associated with different tooth sizes in mouse and hamster. It was recently shown how tooth development scales to species size, notably through the IGF-1 pathway[59]. Consistently, we found that the IGF-1 pathway is over-represented among the co-evolving genes and IGF-1 showed clearly different profiles between mouse and hamster, with co-evolution of the two teeth (Supplementary Fig. 4). But this coevolution also concerns developmental innovations which seem required for upper molar adaptation, the increased bucco-lingual axis polarization associated with long 1-SEK stage and the local SEK inhibition.

By reanalysing published data[60] we found similar patterns of transcriptome coevolution in bat limbs and wings (Fig. 7), and point that some developmental innovations thought to be causative for the new wing morphology also co-evolved in the posterior limb.

We see many parallels between the two systems. The developmental innovations are either fully shared by the two organs (local SEK inhibition in molars, new expression domain of *Fgf8* and *Grem1* in early developing wing and limb) or attenuated in the co-evolving organ (bucco-lingual polarization). Furthermore, they come with specific developmental innovations (higher mesenchyme:epithelium ratio; late maintenance of *Fgf8* and *Grem1* expression in wing), and interact with them in a non-linear manner characterized by threshold effects. In the lower molar, the threshold for making supplementary lingual cusps is passed when supplementing tooth cultures with Activin βA[16]. In mouse limbs, the threshold to block apoptosis needs the combinatory action of FGF and anti-BMP activity[44] (Fig. 7b).

Previous studies of adaptations in serial organs have associated organ-specific evolution to modular cis-regulatory regions permitting organ-specific expression[61–66]. Because they often use another serial organ as a control, they focused on gene expression innovations that are organ-specific but overlooked those co-evolving in other organs. Yet our results suggest that organ-specific developmental innovations are necessary but not sufficient, and co-evolving innovations help pass the threshold for phenotypic change. Future work should concentrate on identifying the underlying mutations to confirm their essential role in organ-specific morphological evolution.

Finally, these findings have implications for our conceptualization of Developmental System Drift, concerted evolution and pleiotropy. Mouse lower molar and bat limb underwent strong divergence in gene expression and development as compared to their low divergence in morphology, an incongruence called Developmental System Drift (DSD)[67–69]. There is now accumulating evidence that such cryptic changes in developmental systems are frequent in evolution[70–73]. Because natural selection mainly acts on the final product of development, drift in development is neutral with respect to natural selection and divergent developmental paths may be taken to reach similar final phenotypes. Further taking into account that genomes mutate constantly and randomly, DSD appears as a likely alternative to developmental conservation[74,75].

In the present situation, the term drift may be a little confusing since at least part of lower molar and hindlimb development evolution is not random drift: it is concerted with developmental innovation in the other organ, and therefore likely induced by the adaptation of this other organ with mutation displaying pleiotropic effects on lower and upper molar development.

Pavlicev, Wagner, and Felix have already proposed that pleiotropy in the organism may favor DSD of species[72,76]. Such link has been observed in experiments of in silico evolution[77,78] but lack support from empirical data. Wagner and colleagues have also already suggested a link between pleiotropy and concerted transcriptomic evolution. In most multispecies transcriptomic analyses, samples of different organs tend to group by species (so-called "species signal" as observed Fig. 1c), whether they are adult tissues[79] or individual embryonic timepoints[80,81]. This pattern, which had first received little attention, was recently reinterpreted as a conspicuous concerted evolution, possibly driven by the pleiotropy of gene networks, repeatedly used in different organs[81,82]. Our study takes this model a step further by bridging concerted transcriptomic evolution with concerted evolution of developmental mechanisms, and quantifying developmental evolution. We also propose concrete mechanisms for how selection could retain pleiotropic mutations which impacted the development of both molars but left the morphology of lower molars relatively intact. Using phylogenomics to identify the causative mutations in our well understood model-system may bring a strong empirical support to this pleiotropy-DSD model in the future.

## Methods
### Data analysis
R scripts corresponding to the main methods and processed data are available on GitHub (https://github.com/msemon/DriftHamsterMouse).

### Rodent breeding and embryo sampling
CD1 (CD1) adult mice and RjHan:AURA adult hamsters were purchased from Charles River (Italy) and Janvier (France) respectively. Females were mated overnight and the noon after morning detection of a vaginal plug or sperm, respectively, was indicated as ED0.5. Other breeding pairs were kept in a light-dark reversed cycle (12:00 midnight), so that the next day at 16:00 was considered as ED1.0.

Pregnant mouse females were killed by cervical dislocation. Hamster females were deeply anesthetized with a ketamine-xylasine mix administered intraperitoneally before being killed with pentobarbital administered intracardially. All embryos were harvested and thereby anesthetized on cooled Hank's or DMEM advanced medium, weighted on a precision balance after excess of liquid was removed with a whatmann paper, as described in ref. 83 and immediately decapitated.

This study was performed in strict accordance with the European guidelines 2010/63/UE and was approved by the Animal Experimentation Ethics Committee CECCAPP (Lyon, France).

### Estimating embryonic age from embryo weight
Embryo weight is well correlated with developmental age, allowing us to use it as a proxy in mouse and hamster, following[84]. We fitted age of development according to weight (in mg) for hamster and mouse data separately, based on 1047 mouse embryos and 636 hamster embryos respectively, collected over more than 15 years of research. We fitted generalised additive models (GAM) to the data after Box-Cox transformation of weight (libraries mgv version 1.8–35 for GAM and MASS 7.3-53.1 for Box-Cox). These models were preferred to log transformations and linear models, because they allow to treat the data

homogeneously between species, and because the relationship is not perfectly linear between weight and age (Supplementary Fig. 1). These models were then used to predict developmental age, based on weight, for all samples used in this study (RNA-seq analysis, cusp patterning analysis, and in situ hybridizations for several genes).

## Epithelium dissociations and in situ hybridizations

Complete or hemi mandibles and maxillae were dissected in Hank's medium and treated with Dispase (Roche) 10 mg/ml in Hepes/KOH 50 mM ph7.7; NaCl 150 mM at 37 °C for 30 min to 1 h depending on embryonic stage. Epithelium and mesenchyme were carefully separated and fixed overnight in PFA 4% at 4 °C. DIG RNA antisense mouse *Fgf4* and *Shh* probes were prepared from plasmids described elsewhere[85,86]. Mouse *Dkk1*, *Wif1*, Mouse and hamster *Bmp4* probes were newly cloned following RT-PCR or DNA synthesis (Table S1). In situ hybridizations were done according to a standard protocol (DIG mix, DIG antibody and BM purple were purchased from ROCHE). Photographs were taken on a Leica M205FA stereomicroscope with a Leica DFC450 digital camera (Wetzlar, Germany) or on a Zeiss LUMAR stereomicroscope with a CCD CoolSNAP camera (PLATIM, Lyon, France).

## 3D reconstruction and epithelium:mesenchyme ratio quantification

Carefully dissected M1 tooth germs (as for RNA-seq preparation) were fixed overnight in PFA 4%, dehydrated in methanol series and kept in 100% methanol at −20 °C. Following rehydratation, samples were bleached in 3% H202, 0.5% KOH, 1x PBS for 30 min at RT. They were then treated with MACS clearing kit from Miltenyi following manufacturer's instructions, with 5 h permeabilisation and 72 h incubations at 4 °C with primary (Mouse-P cadherin (goat) antibody, R&D Systems, #AF761, 1/200) and secondary (Alexa Fluor 488 Donkey anti-goat antibody, Jacson Immunoresearch #705-545-147, 1:200) antibodies. Nuclear staining was performed with TO-PRO3 (Invitrogen) at 1:1000 in 1x PBS, 0,5% Tween20 for 1 h. Samples were mounted in NEEO agarose 1.5% (ROTH, #2267.2), cleared according to manufacturer instructions, and imaged with a 12x objective on a BLAZE microscope (Miltenyi, step size of 2 micrometers for a lightsheet of 4 micrometers). 3D reconstruction was performed with Imaris 9.8.0 (Bitplane). The epithelial compartment was segmented semi-manually as a "surface" from the cadherin channel with the magic wand tool. The whole sample (epithelium+mesenchyme) was automatically segmented as a "surface" including all nuclei from the to-pro3 channel. The volumes of the corresponding surfaces were extracted from the statistics computed by the software for each surface

## Modelling and comparing cusp patterning dynamics

To compare the dynamic of crown morphogenesis in four teeth (lower and upper molars in hamster and mouse) we need to establish the sequence of primary and secondary signalling centres formation (respectively, PEK and SEK). In mouse, this could be done with time lapse imaging of fluorescent lines[15]. To integrate non-model species like hamster, we had to set up a new method that infers the dynamic based on fixed embryos. We hybridised developing molars against a *Fgf4* probe to reveal PEK and SEKs. The patterns we observed among samples are consistent with a stereotypic and specific sequence of SEK patterning in each tooth and species (Fig. 1b, schemas on the sides). We name each stage by the number of signalling centres (PEK stage then 1-SEK stage, 2-SEK stage etc). PEK stage was defined as a stage with an elongated *Fgf4* signal, and 1-SEK was defined as a stage with a more roundish signal or intermediate signal, which can be more or less deported on the buccal side (see Fig. 5).

Cusp patterning can be seen as a succession of irreversible stages representing step-wise cusp additions. Transition rates between these stages were modelled through continuous time Markov modelling as in ref. 10. The rationale is that if sampling is uniform over the time course of tooth morphogenesis, stages that are rarely sampled are very transient (with high exit rate), while stages that are often sampled last for a longer period of time. In continuous Markov models, the duration of each state follows an exponential distribution, which is not realistic for the stage lengths. So, to have a more realistic stage length distribution, each stage was modelled by several consecutive states, so that its length followed an Erlang distribution, which has a mode different from zero. We built independent models for each species and tooth types. Models are estimated on 121 embryos for the hamster lower molars, 113 for hamster upper, 217 for mouse lower, 187 for mouse upper.

We estimated the duration of each stage in a complete model, with different transition rates for all stages. We also fitted several simpler, nested models, with constraints on the number of different transition rates, up to the most simple model with the same transition rate for all stages. We retained models with three different rates in mouse, and two different rates in hamster, by comparing the fit of the models by likelihood ratio tests in each tooth. Markov models were built by custom scripts calling on R libraries maxLik and expm (maxLik_1.4-8 and expm_0.999-6).

## RNA-seq sample preparation

A total of 32 samples per species, coming from eight individuals, were prepared for the time serie RNA-seq analysis, representing consecutive stages in mouse (ED14.5, 15.0, 15.5, 16.0, 16.5, 17.0, 17.5, 18.0) and nine stages in hamster (ED11.8, 12.0, 12.2, 12.5, 13.0, 13.25, 13.5, 13.75, 14.0). Each sample contained two whole tooth germs, the left and right first molars (M1) of the same female individual, and for a given stage, the upper and lower samples came from the same individual. Harvesting and dissection were performed in a minimal amount of time in advanced DMEM medium. The M1 lower and upper germs were dissected under a stereomicroscope and stored in 200 uL of RNA later (SIGMA). Similarly dissected tooth germs from the same litter and same weight were fixed overnight in PFA 4% for immunolocalization and 3D reconstruction, to check for dissection leaving almost no non-tooth tissue. Examples of dissection are visible in ref. 10. Another embryo of the same litter and same weight was processed as indicated above for *Fgf4* in situ hybridization to check the exact developmental stage. Total RNA was prepared using the RNeasy micro kit from QIAGEN following lysis with a Precellys homogenizer. RNA integrity was controlled on a Bioanalyzer (Agilent Technologies, a RIN of 10 was reached for all samples used in this study). PolyA+ libraries of the large-scale dataset were prepared with the Truseq V2 kit (Illumina, non stranded protocol), starting with 150 ng total RNA and reducing the amplification step to only 12 cycles and sequenced on an Illumina Hiseq2000 sequencer (100 bp paired end reads) at the GENOSCOPE (Evry, France).

For the bucco-lingual dataset, we dissected the 4 first molars (left/right, lower/upper) from a unique mouse E15.0 embryo (weight: 359 mg) as above, except that tooth germs were cut in two halves to isolate buccal and lingual side. Replicates were thus obtained by comparing the right and left side of this same embryo. Total RNAs were extracted and libraries were prepared as above, starting with 50–70 ng total RNAs, where an equal amount of AmbionR ERCC RNA Spike-In Mix1 had been added according to the AmbionR protocol (e.g. 1 μL og a 1 :1000 dilution for each tube). A total of 8 libraries were sequenced (50 bp single-end reads) by the Genomeast Sequencing platform, a member of the France Genomique program.

For the epithelium-mesenchyme dataset, lower and upper mouse and hamster first molars were dissected as above and treated for 15 minutes at 37 °C with Dispase (Roche) 10 mg/ml in Hepes/KOH 50 mM ph7.7; NaCl 150 mM to separate the epithelial from mesenchymal parts which were stored in RNAlater. For the mouse data, we generated samples for 2 stages in 2 replicates, using embryos from the

same litter (stage 15.0 dpc, weight: 350 and 370 mg; stage 16.5 dpc: weight: 788 and 808 mg). Left and right epithelium or mesenchyme were pooled. For the hamster data, we generated samples for a single stage without replication. We pooled the left epithelial or mesenchymal parts from 2 embryos from the same 12.5 dpc litter (413 and 427 mg). A total of respectively 16 and 4 libraries were generated with Truseq V2 kit and sequenced (50 bp single-end reads) by the Genomeast Sequencing platform.

## Milestones to define homologous time window for the two species

We focused on lower molar development to define an homologous time window for mouse and hamster tooth morphogenesis, which was then used to align the RNA-seq and cusp patterning timeseries of the 4 molars in the same way. Based on pictures of dissociated epithelia harvested along with the RNA-seq samples and stained with *Fgf4* see above), we could recognize a typical early cap stage in lower molar samples aged 14.6 for mouse and 12.3 for hamster, which defined an early milestone (relative developmental time 0). Lower molar samples aged 18.0 for mouse and 14.6 for hamster showed *Fgf4* expression in forming crests, which defined a late milestone (relative developmental time 10).

## Multivariate analyses

Multivariate analyses were performed using the ade4 package (ade4_1.7-18[87];). We performed principal component analyses on normalised counts (DESeq basemeans), and between groups analyses on the resulting components, which allowed us to quantify the proportion of variance associated with each factor.

## Expression levels estimation using RNA-seq and differential expression analysis

For the whole tooth germ data (64 samples) we obtained 100 bp paired-end sequences, with on average 46.2 M (millions) reads per sample. For epithelium/mesenchyme and bucco/lingual data (respectively 20 and 8 samples), we obtained 50 bp single-end sequences, with on average 93.7 M and 48.6 M reads per sample respectively. Raw data are publically available in ENA with project accession number: PRJEB52633.

These reads were mapped by using Kallisto (version 0.44.0[88],) to custom reference sequences for hamster and mouse transcriptomes. To generate them, we retrieved mouse and hamster cDNAs from Ensembl (release 93, July 2018, assemblies GRCm38.p6 and MesAur1.0[89]), selected 14536 pairs of one-to-one orthologous transcripts, realigned pairs of sequences with Macse (macse_v2.01[90],) and cropped the alignments to get orthologous segments by using custom scripts to make expression levels comparable between species.

Differential gene expression analysis (DE analysis) was performed on smoothed expression profiles over relative developmental age. Developmental age was estimated with embryo weight (GAM models above). The two milestones defined above (see "milestones" section) were used to convert days of development into relative development age (0-10). The relevance of this choice was confirmed by PCA analysis of the transcriptome data (Fig. 1 and Supplementary Fig. 1)

Expression profiles were fitted by third degree polynomial splines with 2 interior knots, for each tooth and species (bs function of spline R package[91], independently or jointly within tooth and/or species, as explained below. Nested models were tested by DEseq2[92] and the best model was chosen for each gene by comparing the fit of these nested models (FDR adjusted *p*-value < 0.05 from DESeq2 LRT tests). When we compared temporal profiles between species, we accounted for the average level of expression in each species. This is to focus on changes in regulation over development, and to discard potential remaining artifacts in species-specific quantifications. Several tests were performed and are described below with the corresponding figure number.

To compute the distance between pairs of temporal expression profiles (Related to Fig. 2a), we fitted a "complex" model with one specific curve per tooth. We computed for each gene the values predicted by each tooth model for 100 equally distributed points (i) on the timeline, split into 10 time windows and measured the Euclidean distance point by point.

To model the divergence of temporal expression profile in each tooth type separately (Related to Fig. 4d), we compared a "non-divergent" model with a single curve to fit both time series (with a species-specific offset to only consider the temporal dynamic), to a "divergent" model with one specific curve per species (with a species-specific offset).

Selection of the temporal expression profile in the 4 tooth types was done as follows (related to Fig. 4a). The "simple" model fits a single curve for the four time series. The "complex" model fits 4 different curves, one per tooth type. The "hamster/mouse" model has 2 different curves, one per species. The "upper/lower" model has one curve per tooth, including the species-specific offset. The best model was selected for each gene by using a bottom-up approach with the results of four independent tests: t1 compares "hamster/mouse"*versus* simple model; t2: "upper/lower" *versus* simple; t3: complex *vs* "upper/lower"; t4: complex *vs* "hamster/mouse". If t1 and t2 are not significant, then the simple model is chosen. If t1 is significant and not t2, the gene is assigned to: "hamster/mouse". If t2 but not t1: "upper/lower". Finally, if "lower/upper" or "hamster/mouse" and t4: complex.

From this selection procedure, percentage of coevolution among genes was computed as the proportion of "hamster/mouse" models among the selected models as follows (related to Fig. 4a): "hamster/mouse"/("hamster/mouse"+"upper/lower"+"complex").

We then computed the intersection of the results with several lists of genes important for tooth development: 259 genes from the bite-it database (retrieved in July 2019), 187 genes with a phenotype in tooth development (100 "dispensable" genes, 87 "keystone" genes[37],), and 295 genes belonging to 8 pathways active in tooth development (17 genes in ACTIVIN pathway, 81 in BMP, 10 in EDA, 69 in FGF, 32 in SHH, 9 in NOTCH, 11 in TGFB, 96 in WNT, courtesy Jukka Jernvall).

## Functional enrichment

We selected the coevolving genes and used the R packages cluster-Profiler (version 4.10.1), enrichplot (1.22.0) and ReactomePA (1.46.0) packages to quantify functional enrichment in biological processes and in reactome pathways[93,94].

## Measure of pathway activation in RNAseq samples

ROMA was used to quantify activation of WNT, BMP and SHH pathways in the bucco-lingual samples (version rRoma_0.0.4.2000, https://github.com/Albluca/rRoma and[31]). ROMA is designed to compare pathway activity in transcriptomic samples based on expression levels of a list of targets for the pathway. Genes for the SHH modules were retrieved from GSEA ([95], 41 genes present in our dataset). Because BMP and WNT pathways are active both in the mesenchyme and the epithelium and they target different genes in each tissue[96], we used two separate lists of targets to estimate both an epithelial and a mesenchymal activity, adapted from a "regulatory evidence" dataset established for first lower molar development[96]. Building on literature and their own transcriptomic analysis, the authors had defined target genes based on their up or downregulation following activation or inactivation of each pathway. For data consistency, we selected only targets established in the study from transcriptome analysis, in 13.5 and 14.5 dpc epithelium and 10.5 dpc mesenchyme. Different modules were built for activities in the mesenchyme and epithelium compartments. For BMP in the mesenchyme, we considered 15 genes as positive targets and 4 as negative targets (further noted 15:4). In the epithelium, the numbers of positive:negative targets were respectively 32:34. For WNT, we built modules with 4:31 positive:negative targets in

the mesenchyme, and 33:45 in the epithelium. These in-silico quantifications were consistent with many known aspects of tooth development. Buccal compartments all show high levels of pathway activity, consistent with the presence of the first SEK acting as a source of WNT, BMP and SHH signals. Lingual compartments show much lower levels of signalling activities than buccal compartments, consistent with their distance to the first SEK. The lower lingual compartments show BMP and WNT activities that are higher in epithelium than in mesenchyme, consistent with the fact that epithelial activation predates mesenchymal activation in tooth development.

### Estimating tissue proportions from RNA-seq data by deconvolutions

We used the R package DeconRNASeq (DeconRNASeq_1.32.0[36]) to estimate the relative proportions of epithelium and mesenchyme compartments in bulk tooth germ transcriptomes. We defined lists of marker genes for each compartment by pairwise differential analysis of tissue-specific transcriptomes (DESeq2, log2 fold change > 3, adjusted $p$-value < 0.05). We used 1025 mesenchyme and 621 epithelium marker genes found by comparing 10 epithelium and 10 mesenchyme RNAseq samples, mixing tooth, stages and species. We estimated the accuracy of the prediction by bootstrapping 1000 times the marker lists. The relative proportions of buccal and lingual compartments was inferred by a similar procedure. We used 414 buccal and 235 lingual marker genes, from the differential analysis of 8 samples (DESeq2, log2 fold change > 1, adjusted $p$-value < 0.05).

### Expression levels and transcriptome dynamics in bats

We downloaded all bats and mouse raw RNA-seq samples from a published dataset (SRP061644, Maier et al. 2017), totalizing 17 samples in mouse and 16 in bat (*Carollia Perspicillata*) at three consecutive stages: ridge (E10.0 for mouse; CS13 for bat), bud (E10.5; CS14) and paddle (E11.5; CS15) stage. Bat reads were assembled de novo with Trinity v2.14.0[97], by using single end mode and in silico normalisation. Bat expression levels were quantified by Salmon[98] with the script provided by Trinity (align_and_estimate_abundance.pl). Mouse reads were directly mapped with Salmon to the GENCODE mouse transcriptome reference (gencode.vM29.pc[99],). Bat transcripts were assigned to mouse orthologs by blastn[100]. Blast and Trinity were run with prebuilt dockers. Differential analysis was run over smoothed expression profiles like in the method section "Expression levels estimation using RNA-seq and differential expression analysis". Code is available here: https://github.com/msemon/DriftHamsterMouse

### Reporting summary

Further information on research design is available in the Nature Portfolio Reporting Summary linked to this article.

## Data availability

Raw data are publicly available in ENA with project accession number: PRJEB52633. For more conveniency, see also Supplementary Data 1 and Supplementary Tables 1 and 2. Source data are provided with this paper.

## Code availability

All custom code (run in R) used in this study is made available for each figure panel, together with the accompanying data, so that all the panels can be reproduced. They are part of the source file, a zip containing a file per figure, with code and data. See also https://github.com/msemon/DriftHamsterMouse.

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

## Acknowledgements

We acknowledge the contribution of several platforms of SFR Biosciences Gerland-Lyon Sud (UMS344/US8): the Plateau de Biologie Expérimentale de la Souris (PBES) (many thanks especially to Jean-Louis Thoumas, Tiphaine Dorel, Céline Angleraux, Marie Teixeira, Myriam Prudent), the Plateau Technique Imagerie/Microscopie (PLATIM) as well as the computer resources from PSMN (ENS Lyon). We acknowledge the technical help of Anne Lambert, Alain Rubod, Mathilde Estevez-Villar, and the contribution of many students including Coraline Petit, Alice Lorenc, Margaux Pillon, Ludivine Rotard and Asma Benahmed. We are grateful to several colleagues and their staff for sending plasmid probes: Irma Thesleff (*Fgf4*), Hiko Ogura (*Cv2/Bmper*), D. Duboule (*Fgf10*). We kindly thank Joanne Burden, Mirko Francesconi, Marie Delattre, Michaelis Averof, Pascal Hagolani, and Jukka Jernvall and Pierfrancesco Pagella for their feedback on the manuscript. This work was supported by the Agence Nationale pour la Recherche (ANR 2011 JSV6 00501 "Convergdent"), a grant from the GENOSCOPE - Centre National de Séquençage, a grant from IDEX Lyon ANR-16-IDEX-0005, and an European Council Research grant (ERC 2022 COG PLEIOTROPY 101088398). Salaries were supported by the Centre National de la Recherche Scientifique, the Ecole Normale Supérieure de Lyon and the Université de Lyon, Université Lyon 1.

## Author contributions

S.P. and M.S. conceived the study, designed and performed the experiments, analysed the data and wrote the manuscript. M.M., M.P., C.C. and S.P. collected embryos and performed in situ hybridizations. C.C. performed immunostaining and 3D reconstructions. Models and code for cusp patterning were developed and refined by L.G., and models of temporal expression were initiated by P.V. All authors read and commented on the manuscript.

## Competing interests

The authors declare no competing interests.
