## [Transparent Peer Review file · Nature Communications]

Comparative transcriptomics in serial organs uncovers early and pan-organ developmental changes associated with organ-specific morphological adaptation

Corresponding Author: Dr Sophie Pantalacci

Version 0:

Reviewer comments:

Reviewer #1

(Remarks to the Author)

The manuscript by Semon et al. provides an impressively thorough analyses of the development of mouse and hamster molars, with some additional data on the limb development in mice and bats. The comparisons between and within species are designed such that one can compare the expression profiles of derived morphologies (upper molars in mice, fore limbs in bats) to that of underived ones (hamster upper molars and mouse fore limbs), as also have the counter parts within the species that show less evolutionary disparity (hind limbs and lower teeth). Overall, the authors argue for a mixture of coevolution of expression profiles between developmental systems (e.g. upper and lower molars, irrespective of morphology), expression changes linked to derived morphologies (e.g. mouse upper molars), and also evidence for extensive developmental system drift (DSD) suggesting evolving network dynamics irrespective of morphology. I find the analyses very interesting, and data to be of high quality and credible. Currently I also find these very exciting analyses to be a bit muddled by the writing and lack of clarity in conceptual issues related to organ development. All these should be easily fixable and below I first explain the conceptual issue, followed by specific comments about the analyses, text and figures.

Conceptual comments (my apologies for this overtly extensive discussion):

The authors seem to build their conceptual framework around rather hard core, Haeckelian 'recapitulation' theory, and heterochrony. Considering the extensive literature on the general topics, this seems rather disconnected to the actual study. Importantly, the order of evolution of new tooth cusps and their developmental origins are well established. The key developmental principles being that taller cusps are initiated first, and shorter cusps later during ontogeny of an individual tooth. This follows generally the evolutionary order of acquisition of new cusps as they start small, and then evolve taller in many lineages. The mouse and hamster molars are examples where the main cusps have evolved to be almost equal in height, but still retain roughly the order of evolutionary origins in their developmental initiation. So yes, the initiation of cusp development, or patterning in general, kind of recapitulates the evolutionary history, but this order is not set in stone as there are exceptions. Most notably when the metacone cusp is taller than the paracone, the metacone is also developing earlier than the evolutionary older paracone. This also means that new cusps, not present in the ancestors are not result of recapitulation, but rather a product of the patterning cascade mode of cusp development (see below).

Overall, the beautiful results on mouse and hamster cusp initiation the authors observe fit well what is known about the cusp development. One of results that the authors seem to be surprised about is the fact that the evolutionary new third-row of cusps in mouse upper molar appears to stem from early polarization in the lateral (lingual) expression dynamics. Hence, the developmental 'decision' to expand lingually to make room for the third cusp row seems to be made much earlier in tooth development. This is a very exciting result, and it fits also the dynamic and iterative nature of tooth development. Namely, the patterning cascade model of tooth cusp development where iterative activation of secondary enamel knots initiate cusp development postulates that developmental differences (such as changes in tissue growth rates or inhibition of new enamel knots) have cumulative effects such that the later developing, short cusps can be generated whereas the earlier developing, tall cusps show relatively little change. Thus, to make the third row of cusps in the mouse would require changes earlier on in the crown formation, exactly what the authors have shown.

What I suggest is that the authors read the works below and examine whether they could make their analyses to link to the literature in more concrete terms. This would make the study more mechanistic and stimulate more studies down the road (and more cited!).

For works on the basics of tooth cusp evolution and development, Percy Butler is still highly relevant, including classics such as:

Butler, P. M. 1995: Ontogenetic aspects of dental evolution. *Int. J. Dev. Biol.* 39: 25-34.

Butler, P. M. 1956: The Ontogeny of Molar Pattern. *Biol. Rev.* 31: 30-70.

The issue of cusp initiation order has been examined in species where the phylogenetically old cusps is not the tallest:

Berkovitz, B. K. B. 1967: The dentition of a 25-day pouch-young specimen of *Didelphis virginiana* (Didelphidae: Marsupialia). *Arch. Oral Biol.* 12: 1211-1212.

Also Yamanaka is investigating this in the house shrew, and at least the nice review mentions the development of the upper molar cusps:

Yamanaka: Evolution and development of the mammalian multicuspid teeth *Journal of Oral Biosciences* 64 (2022) 165e175

Because the secondary enamel knots are used as the readout of the patterning, it might be useful to check my paper discussing enamel knots and patterning cascade mode of development where small changes active over the whole tooth development can have cumulative effects on later developing cusp. Computational models (e.g. ToothMaker) were built on this logic.

Jernvall, J. Linking development with generation of novelty in mammalian teeth. *PNAS, USA* 97: 2641-2645 (2000).

Specific comments:

1) The analyses of limbs and bats are brought up in the end of the introduction, almost casually. It would make the work a bit more balanced if the part on limbs is brought up more clearly. This could be as little as saying in the end of the introduction something like 'Third, we further tested the patterns observed in teeth in an another much studied system, the mammalian limb with a special focus on bat wing. This system offers...and so on.

2) The authors stress that the lower molars of the hamster and the mouse are rather alike. This will certainly make rodentologist cringe a bit because, compared to the lower molars of the mouse, hamsters have lateral offset of cusps, the cusps are erect, there is a central crest joining cusps mesiodistally, the anteroconid is more separate and symmetrical, and the enamel distribution is more uniform. I am not insisting that the premise of the work is wrong, as the main focus is on cusp number, and the presence of the third cusp row in mouse upper molars. Nevertheless, it would be prudent to mention that some of the species difference in expression profiles between lower molars could be due to the differences in these details of crown shape. Indeed, as features such as sloping of cusps and presence/absence of longitudinal links are shared between the upper and lower molars within species, these could also explain partially the coevolution of the expression profiles between upper and lower molars.

3) Yet another difference between the species is that golden hamsters molars are a fair bit of larger than mouse molars. Presumably hamster molars also grow faster, and the patterning appears to be fast too (Fig. 1B). As IGF signaling has been implicated in molar scaling, could genes linked to this pathway explain some of the differences between the species? I leave it to the authors to decide whether to mention or not.

4) The rather extensive developmental system drift (DSD inferred for the expression dynamics between hamster and mouse lower molars is very interesting. I think this is a rather general result in that the more phylogenetically distant species are compared, the more we see differences (even though at the level of keystone gene categories the patterns can be conserved). This makes one wonder why F1 hybrids between species are typically intermediate, even when they are phylogenetically quite distant and phenotypically disparate? Should rampant DSD produce weird hybrids? I do not have an answer to these questions, and do not expect the authors to address this. It is just one of the stimulating inference that the work makes one think about.

5) There are quite a few parts in the text that would benefit rewording, some of the listed below.

Page 2

"The principle of recapitulation stipulates that successive evolutionary steps (phylogeny) are replayed in successive developmental steps (ontogeny)(Abzhanov, 2013; Gould, n.d.; Uesaka & Irie, 2022)."

COMMENT: Recapitulation is not a central concept in current literature, or null expectation how evolutionary changes are made. Even just rewording to "Historically, the principle of recapitulation stipulated...." would help.

Page 3

"Here we combine in situ marker-gene with bulk transcriptome time series to ask how developmental trajectory evolves when a new shape evolves, in an iconic evo-devo model, the rodent molar."

COMMENT: Although this referee fully sympathizes with the 'iconic' status of rodent molars in evodevo, perhaps better say 'widely used'

"Between 18-12 million years ago (MYA), the upper molars of mouse ancestors gradually acquired two supplementary cusps on the lingual side, and reduced cusps size on the buccal side (Figure 1A)."

COMMENT: The Fig. 1A is showing something very different. But it would be useful to have a figure as stated.

"This new dental plan and accompanying changes in mastication movements are adaptive and associated with the success of murine rodent radiation(Lazzari et al., 2008; Tiphaine et al., 2013)."

COMMENT: Perhaps better state something like: This new occlusal morphology has been linked to changes in mastication

and new dietary adaptations, facilitating the success of murine rodent radiation (Lazzari et al., 2008; Tiphaine et al., 2013).

Page 16

"We found that Bmp4, a known inhibitor (Meguro et al., 2019)"

COMMENT: Mesenchymal Bmp4 is required for the induction of enamel knot, but it is then upregulated in the enamel knots themselves during their apoptotic silencing. Meguro et al 2019 used K14-Bmp4 mice, hence they upregulated the epithelial expression.

Page 21

"Second, more surprisingly, we revealed the marked developmental co-evolution of the lower molar, which we had first introduced as a control."

COMMENT: See my comment 2) above. Surprisingly is a rather strong word to be used about own results. Strikingly?

Page 22

"Finally, supplementary cusps in mouse mutants are limited to small accessory cusps,"

COMMENT: Typically limited to small cusps? There are exceptions, e.g., Sostdc1.

Page 23

"The latest steps of ontogeny are modified during evolution to produce novelty, so that ontogeny recapitulates phylogeny."

"Our integrated study of the whole developmental sequence however points to a very different hidden scenario (Figure 6C)."

"Thus, upper molar evolution is a case of recapitulation but not a case of terminal addition."

COMMENT: See my first (very long) comment. The first sentence above sounds incorrect as written because, for example, the third cusp row in upper mouse molars is a novelty not present in the ancestors. How can you recapitulate phylogeny to produce something that did not exist in the ancestors? The second sentence links to the third that I find not very tangible (see my first very long comment). That is, the addition of new cusps such as the third cusp row can be considered a terminal addition in a sense that new, small cusps emerge late in the tooth ontogeny. But as the authors have very correctly documented, the actual process that produces these novelties starts earlier in tooth development, visible in their gene expression dynamics. This is because you have a patterning cascade of iterative signaling between the tissues and signaling center. A very exciting result with no need to invoke the can of recapitulation.

6) Figures

These need some work. Fig 1A has the tooth on the right pointing different direction as the ones on the left. Use 'Upper' and 'Lower' (not Up and Low). Many figures are very complex. See if can make them simpler, or explain the point in the figure title.

Reviewer #2

(Remarks to the Author)

The authors pursue a fundamental question in developmental and evolutionary biology, namely the evolution of morphogenesis and whether this is fundamentally due to alternations in the final phases of organs development, or it is already coded from earlier moments. They do so with an approach that I personally consider extremely powerful - comparing the morphogenetic process across multiple levels. To do so, they choose a perfectly suitable model, the rodent molar: molars are teeth that are highly homologous during vertebrate evolution, and their morphogenesis is strictly dependent on the local genetic programs, cell-cell interactions and physical cues, while it is mostly independent from environmental influences and from the influence of other bodily structures such as vasculature, systemic signals, or others (all factors that acquire importance at later stages, e.g. during mineralization).

The work is original, robust, timely, well-conducted, and clearly exposed. I believe only minor aspects should be addressed prior to publication:

- The quantification of the epithelium:mesenchyme ratio via transcriptome deconvolution is smart, yet it would be more convincing to use more direct methods – cell counting / tissue measurement (weight/volume).

- The analysis of the co-evolution of gene expression profiles is extremely interesting, yet it is represented and described very "globally" both in the text and in figure 4. I suggest to the authors to represent in more detail and more openly the gene networks whose expression co-evolved – if space is not sufficient, they could choose those whose co-evolution score is higher. This could lead also to a more thorough discussion of how concerted evolution of genomic networks lead to major morphological changes.

- The observation that several developmental phenotypes of the lower molar have evolved in concert with the upper molar without driving major phenotypic changes is really important. Did the authors investigate whether some of the changes between mouse and hamster's lower molar transcriptome could, apparently paradoxically, have had an inhibitory effect on the generation of phenotypic changes? In other words, is it possible that the lack of major phenotypic changes in the lower

molar is caused by the evolution of compensatory genetic networks?

- The analysis of co-evolution of transcriptomes during bat limb evolution would really benefit from a more extensive description and a more complete figure. I am fully aware that the article uses mostly the tooth as model and extends to the bat limb mostly to assess whether the principle identified during tooth evolution applies to other organs. Yet, this would make the case of the authors more compelling and more convincingly applicable to other organs (and it would be a pity to bury such an interesting analysis on the bat by using it so summarily)

Pierfrancesco Pagella
Associate Professor, Linköping University

Reviewer #3

(Remarks to the Author)
N/A

Version 1:

Reviewer comments:

Reviewer #1

(Remarks to the Author)
I have now read the revised manuscript and the rebuttal. The authors have done a thorough job in revising the work, and I thank them for addressing my lengthy comments. The figures are much improved, and the text is clearer. When I write clearer, it means that I would still wish the presentation to be less like a podium presentation in a large hall, and more like a meticulous exposition of scientific discovery. But I leave it to the authors to decide how much they want to tinker with the language. The first paragraph of the introduction remains particularly declaratory. But overall, this is a very valuable contribution.

Couple of oddities:

Page 12

“This may root in differences of basal expression levels or this may reflect co-evolution of the temporal dynamics of morphogenesis”

COMMENT: root?

End of the discussions:

“could retain pleiotropic mutations which impacted the development of both molars but poorly changed the lower molar”

COMMENT: Poorly? Maybe ‘but leave the morphology of lower molars relatively intact’ or perhaps ‘conserved’

(Remarks on code availability)

The github page looks all right, and the instructions are clear. I have not run the scripts.

Reviewer #2

(Remarks to the Author)

The authors addressed all my comments. Great work!

(Remarks on code availability)

Point-by-point rebuttal on Semon et al.

Review #1

The manuscript by Semon et al. provides an impressively thorough analyses of the development of mouse and hamster molars, with some additional data on the limb development in mice and bats. The comparisons between and within species are designed such that one can compare the expression profiles of derived morphologies (upper molars in mice, fore limbs in bats) to that of underived ones (hamster upper molars and mouse fore limbs), as also have the counter parts within the species that show less evolutionary disparity (hind limbs and lower teeth). Overall, the authors argue for a mixture of coevolution of expression profiles between developmental systems (e.g. upper and lower molars, irrespective of morphology), expression changes linked to derived morphologies (e.g. mouse upper molars), and also evidence for extensive developmental system drift (DSD) suggesting evolving network dynamics irrespective of morphology. I find the analyses very interesting, and data to be of high quality and credible. Currently I also find these very exciting analyses to be a bit muddled by the writing and lack of clarity in conceptual issues related to organ development. All these should be easily fixable and below I first explain the conceptual issue, followed by specific comments about the analyses, text and figures.

R: We thank [REDACTED] for his positive appreciation of our results and for expressing his reservations in detail. We think we in fact share quite similar views, and this helped reaching a more faithful version of our manuscript.

Conceptual comments (my apologies for this overtly extensive discussion):

The authors seem to build their conceptual framework around rather hard core, Haeckelian 'recapitulation' theory, and heterochrony. Considering the extensive literature on the general topics, this seems rather disconnected to the actual study. Importantly, the order of evolution of new tooth cusps and their developmental origins are well established. The key developmental principles being that taller cusps are initiated first, and shorter cusps later during ontogeny of an individual tooth. This follows generally the evolutionary order of acquisition of new cusps as they start small, and then evolve taller in many lineages. The mouse and hamster molars are examples where the main cusps have evolved to be almost equal in height, but still retain roughly the order of evolutionary origins in their developmental initiation. So yes, the initiation of cusp development, or patterning in general, kind of recapitulates the evolutionary history, but this order is not set in stone as there are exceptions. Most notably when the metacone cusp is taller than the paracone, the metacone is also developing earlier than the evolutionary older paracone. This also means that new cusps, not present in the ancestors are not result of recapitulation, but rather a product of the patterning cascade mode of cusp development (see below).

Overall, the beautiful results on mouse and hamster cusp initiation the authors observe fit well what is known about the cusp development. One of results that the authors seem to be surprised about is the fact that the evolutionary new third-row of cusps in mouse upper molar appears to stem from early polarization in the lateral (lingual) expression dynamics. Hence, the developmental 'decision' to expand lingually to make room for the third cusp row seems to be made much earlier in tooth development. This is a very exciting result, and it fits also the dynamic and iterative nature of tooth development. Namely, the

patterning cascade model of tooth cusp development where iterative activation of secondary enamel knots initiate cusp development postulates that developmental differences (such as changes in tissue growth rates or inhibition of new enamel knots) have cumulative effects such that the later developing, short cusps can be generated whereas the earlier developing, tall cusps show relatively little change. Thus, to make the third row of cusps in the mouse would require changes earlier on in the crown formation, exactly what the authors have shown.

What I suggest is that the authors read the works below and examine whether they could make their analyses to link to the literature in more concrete terms. This would make the study more mechanistic and stimulate more studies down the road (and more cited!).

For works on the basics of tooth cusp evolution and development, Percy Butler is still highly relevant, including classics such as:

Butler, P. M. 1995: Ontogenetic aspects of dental evolution. *Int. J. Dev. Biol.* 39: 25-34.

Butler, P. M. 1956: The Ontogeny of Molar Pattern. *Biol. Rev.* 31: 30-70.

The issue of cusp initiation order has been examined in species where the phylogenetically old cusps is not the tallest:

Berkovitz, B. K. B. 1967: The dentition of a 25-day pouch-young specimen of *Didelphis virginiana* (Didelphidae: Marsupialia). *Arch. Oral Biol.* 12: 1211-1212.

Also Yamanaka is investigating this in the house shrew, and at least the nice review mentions the development of the upper molar cusps:

Yamanaka: Evolution and development of the mammalian multicuspid teeth *Journal of Oral Biosciences* 64 (2022) 165e175

Because the secondary enamel knots are used as the readout of the patterning, it might be useful to check my paper discussing enamel knots and patterning cascade mode of development where small changes active over the whole tooth development can have cumulative effects on later developing cusp. Computational models (e.g. ToothMaker) were built on this logic.

Jernvall, J. Linking development with generation of novelty in mammalian teeth. *PNAS, USA* 97: 2641-2645 (2000).

R: We are very grateful for this feedback, as obviously, our view was insufficiently explained (including the difference between recapitulation or heterochronies taken as observed patterns versus as evolutionary mechanisms), leading to misinterpretation. Actually, we think we fully agree with all the raised points, which we understand as follows.

The deep system's understanding of molars could predict that no terminal addition (both as a pattern and mechanism) would be found there, and this is therefore not a big surprise for those who have this knowledge. This system's understanding was built on a large body of experimental evidence in mouse embryos and the corpus of adult tooth species diversity and formed the logic of the "Toothmaker" model. By providing the first thorough description of the development of two species with different cusp numbers, our results therefore somehow validate this theoretical corpus. Finally, we fully agree that recapitulation occurs because of developmental system's rules, and not according to outdated Haeckel's theory.

We revised extensively the text of our manuscript to take [REDACTED] detailed feedback into account, while keeping a general focus. Indeed, it was not our point to discuss a strict recapitulation law in molars in the frame of this paper, and “recapitulation” was only meant regarding the late addition of the supplementary cusps in development and the fossil record. We will discuss in a separate manuscript the order of cusp formation in lower and upper molars and how it may recapitulate mammalian molar evolution. Also, what appears obvious to the tooth community may not be as clear for other communities, which do not have such an advanced system’s understanding of morphology, and may have other expectations. That’s why we kept a more general focus, while making the following extensive changes:

- the abstract was modified to clearly convey the idea that our study nicely illustrates how the observed patterns (recapitulation, heterochrony) are a simple (predictable) consequence of the logic of developmental mechanisms.
- introduction was nearly completely rewritten, to better introduce the tooth model as a dynamic model system, in which evolutionary patterns have been examined in the light of development since more than one century.
- results: we added references on previous studies on cusp formation (on top of other modifications detailed in the rest of this rebuttal)
- discussion: we better emphasize how our study provides an experimental validation of the morphodynamic model of tooth development, which predicts the recapitulation pattern of the supplementary cusps. We now refer both to Jernvall 2000 and Yamanaka 2023 (thank you for pointing out this interesting review which we had missed). We discuss more explicitly how a recapitulation pattern can be the consequence of the phylogenetic history of a patterning system.

Specific comments:

1) The analyses of limbs and bats are brought up in the end of the introduction, almost casually. It would make the work a bit more balanced if the part on limbs is brought up more clearly. This could be as little as saying in the end of the introduction something like 'Third, we further tested the patterns observed in teeth in another much studied system, the mammalian limb with a special focus on bat wing. This system offers...and so on.

R: This was done. See also the new Figure 6 entirely dedicated to bat analysis.

2) The authors stress that the lower molars of the hamster and the mouse are rather alike. This will certainly make rodentologist cringe a bit because, compared to the lower molars of the mouse, hamsters have lateral offset of cusps, the cusps are erect, there is a central crest joining cusps mesiodistally, the anteroconid is more separate and symmetrical, and the enamel distribution is more uniform. I am not insisting that the premise of the work is wrong, as the main focus is on cusp number, and the presence of the third cusp row in mouse upper molars. Nevertheless, it would be prudent to mention that some of the species difference in expression profiles between lower molars could be due to the differences in these details of crown shape. Indeed, as features such as sloping of cusps and presence/absence of longitudinal links are shared between the upper and lower molars within species, these could also explain partially the coevolution of the expression profiles between upper and lower molars.

R: We agree and put more stress on these differences, both in the introduction and discussion:

Changes in the introduction: "Changes to the mouse lower molar dental plan were much less drastic: cusp number was conserved, and changes were limited to *reducing the lateral offset* and changing connections between cusps to enable the new occlusion. "

Changes in the discussion: " Part of this may be associated with coevolving morphological features (e.g. cusp height *and slope*, *absence of a longitudinal crest linking cusps in mouse*).

3) Yet another difference between the species is that golden hamsters molars are a fair bit of larger than mouse molars. Presumably hamster molars also grow faster, and the patterning appears to be fast too (Fig. 1B). As IGF signaling has been implicated in molar scaling, could genes linked to this pathway explain some of the differences between the species? I leave it to the authors to decide whether to mention or not.

R: We have now extended the functional analysis of the co-evolving genes. IGF1 was part of the enriched pathways, with IGF-1 as a coevolving gene, as shown in the new Supplementary Fig.4. We also included a sentence in the corresponding section of the discussion:

"However, we show that the lower molar development has evolved as much as, and coevolved with, the upper molar (Figure 1D and 4D). Part of this may be associated with coevolving morphological features like cusp height and slope, absence of a longitudinal crest linking cusps in mouse. Another part may be associated with different tooth sizes in mouse and hamster. It was recently shown how tooth development scales to species size, notably through the IGF-1 pathway (Chistensen et al, PNAS 2023). Consistently, we found that the

IGF-1 pathway is over-represented among the co-evolving genes and IGF-1 showed clearly different profiles between mouse and hamster, with co-evolution of the two teeth (Supplementary Fig.4). "

4) The rather extensive developmental system drift (DSD inferred for the expression dynamics between hamster and mouse lower molars is very interesting. I think this is a rather general result in that the more phylogenetically distant species are compared, the more we see differences (even though at the level of keystone gene categories the patterns can be conserved). This makes one wonder why F1 hybrids between species are typically intermediate, even when they are phylogenetically quite distant and phenotypically disparate? Should rampant DSD produce weird hybrids? I do not have an answer to these questions, and do not expect the authors to address this. It is just one of the stimulating inferences that the work makes one think about.

R: Interesting ! Intermediate average but few weird outliers???

5) There are quite a few parts in the text that would benefit rewording, some of the listed below.

Page 2

"The principle of recapitulation stipulates that successive evolutionary steps (phylogeny) are replayed in successive developmental steps (ontogeny (Abzhanov, 2013; Gould, n.d.; Uesaka & Irie, 2022))."

COMMENT: Recapitulation is not a central concept in current literature, or null expectation how evolutionary changes are made. Even just rewording to "Historically, the principle of recapitulation stipulated...." would help.

R: This part was reworded. In particular, we pay attention that the wording focus on recapitulation as a pattern and not a mechanism.

"Current understanding of developmental evolution is still largely influenced by observations made by comparative developmental biologists in the 19th century and revisited in the 1980s by SJ Gould and others. They emphasized the parallels between development and evolution, caricatured by the formula "ontogeny recapitulates phylogeny", according to which successive evolutionary steps can be reflected in successive development steps, and the latest phylogenetic changes take root in the latest stages of development¹⁻³. "

Page 3

"Here we combine in situ marker-gene with bulk transcriptome time series to ask how developmental trajectory evolves when a new shape evolves, in an iconic evo-devo model, the rodent molar."

COMMENT: Although this referee fully sympathizes with the 'iconic' status of rodent molars in evodevo, perhaps better say 'widely used'

R: Changed for "ideal"

"Between 18-12 million years ago (MYA), the upper molars of mouse ancestors gradually acquired two supplementary cusps on the lingual side, and reduced cusps size on the buccal side (Figure 1A)."

COMMENT: The Fig. 1A is showing something very different. But it would be useful to have a figure as stated.

Changed for 1B, where molars are shown.

"This new dental plan and accompanying changes in mastication movements are adaptive and associated with the success of murine rodent radiation(Lazzari et al., 2008; Tiphaine et al., 2013)."

COMMENT: Perhaps better state something like: This new occlusal morphology has been linked to changes in mastication and new dietary adaptations, facilitating the success of murine rodent radiation (Lazzari et al., 2008; Tiphaine et al., 2013).

R: The sentence was replaced.

Page 16

"We found that *Bmp4*, a known inhibitor(Meguro et al., 2019)"

COMMENT: Mesenchymal *Bmp4* is required for the induction of enamel knot, but it is then upregulated in the enamel knots themselves during their apoptotic silencing. Meguroe et al 2019 used K14-*Bmp4* mice, hence they upregulated the epithelial expression.

R: We fully agree with this. We modified the text to make it more obvious that we are focusing on epithelial expression and activity of BMP4 in this part.

We found that *Bmp4*, which acts in the epithelium as a cusp formation inhibitor (Meguro et al, 2019),

Page 21

"Second, more surprisingly, we revealed the marked developmental co-evolution of the lower molar, which we had first introduced as a control."

COMMENT: See my comment 2) above. Surprisingly is a rather strong word to be used about own results. Strikingly?

R: Changed for Strikingly

Page 22

"Finally, supplementary cusps in mouse mutants are limited to small accessory cusps,"

COMMENT: Typically limited to small cusps? There are exceptions, e.g., *Sostdc1*.

R: Changed for typically

"The latest steps of ontogeny are modified during evolution to produce novelty, so that ontogeny recapitulates phylogeny."

"Our integrated study of the whole developmental sequence however points to a very different hidden scenario (Figure 6C). "

"Thus, upper molar evolution is a case of recapitulation but not a case of terminal addition."

COMMENT: See my first (very long) comment. The first sentence above sounds incorrect as written because, for example, the third cusp row in upper mouse molars is a novelty not present in the ancestors. How can you recapitulate phylogeny to produce something that did not exist in the ancestors? The second sentence links to the third that I find not very tangible (see my first very long comment). That is, the addition of new cusps such as the third cusp row can be considered a terminal addition in a sense that new, small cusps emerge late in the tooth ontogeny. But as the authors have very correctly documented, the actual process that produces these novelties starts earlier in tooth development, visible in their gene expression dynamics. This is because you have a patterning cascade of iterative signaling between the tissues and signaling center. A very exciting result with no need to invoke the can of recapitulation.

R: We feel that our point is in fact quite close from Jukka Jenvall's view.

Not knowing about the developmental mechanisms, just from observation at a glance, we may think it is a terminal addition: the two phylogenetically younger cusps are added at the end of the trajectory, in the same order as during evolution (we clarified in the text that our point on recapitulation only concerns the supplementary cusps, and not the whole sequence - which we don't want to discuss there).

Our good knowledge of the morphodynamic nature of tooth morphogenesis however predicts that terminal addition is a very poorly credible mechanism. Changes in the final shape instead root in the full dynamics, and at best should become gradually visible as the trajectory proceeds, cumulating as [REDACTED] suggests.

But we see differences from the earliest step of the trajectory, and the teeth look quantitatively even sharper at these earlier stage, than what they are later. This is not surprising for a morphodynamic mechanism, especially to reach such final differences in pattern and shape. A more gradual difference may produce just accessory cusps.

As said above, the introduction and parts on recapitulation were fully rewritten to better emphasize this view.

6) Figures

These need some work. Fig 1A has the tooth on the right pointing different direction as the ones on the left. Use 'Upper' and 'Lower' (not Up and Low). Many figures are very complex. See if can make them simpler, or explain the point in the figure title.

R : We are aware that the figures are complex. We had already given (and gave some more) thought to try and make it simpler. We also rewrote the figure titles and subtitles in the figure to make them more explanatory. Up and low were changed to Upper and lower throughout the figures and tooth orientation was corrected on Fig1A.

Thank you again for this very constructive comments.

Review #2

The authors pursue a fundamental question in developmental and evolutionary biology, namely the evolution of morphogenesis and whether this is fundamentally due to alternations in the final phases of organs development, or it is already coded from earlier moments. They do so with an approach that I personally consider extremely powerful - comparing the morphogenetic process across multiple levels. To do so, they choose a perfectly suitable model, the rodent molar: molars are teeth that are highly homologues during vertebrate evolution, and their morphogenesis is strictly dependent on the local genetic programs, cell-cell interactions and physical cues, while it is mostly independent from environmental influences and from the influence of other bodily structures such as vasculature, systemic signals, or others (all factors that acquire importance at later stages, e.g. during mineralization).

The work is original, robust, timely, well-conducted, and clearly exposed. I believe only minor aspects should be addressed prior to publication:

R: We thank Pierfrancesco Pagella for his positive evaluation of our manuscript and of our approach. We also thank him for his constructive comments we address below.

- The quantification of the epithelium:mesenchyme ratio via transcriptome deconvolution is smart, yet it would be more convincing to use more direct methods – cell counting / tissue measurement (weight/volume).

R: We thank the reviewer for this praise for the use of deconvolution. To directly measure the volumes of epithelium and mesenchyme we have now implemented a method based on lightsheet imaging to obtain 3D reconstructed tooth germs. This extends in hamster and confirmed in mouse what we had previously done in mouse only with 3D reconstructions based on confocal imaging (Pantalacci et al, Genome Biology 2017). We added the obtained epithelium:mesenchyme ratio to Figure 3 A and Table S2. These volumes are remarkably consistent with the results from deconvolutions (which proves also, to us, the power of this technique).

a. Tooth germ composition mesenchyme vs epithelium

- The analysis of the co-evolution of gene expression profiles is extremely interesting, yet it is represented and described very “globally” both in the text and in figure 4. I suggest to the authors to represent in more detail and more openly the gene networks whose expression co-evolved – if space is not sufficient, they could choose those whose co-evolution score is higher. This could lead also to a more thorough discussion of how concerted evolution of genomic networks lead to major morphological changes.

R: This suggestion led us to three different analyses, two of which are now presented in Figure 4A and in Supplementary Figure 4 and the third one is detailed below.

1. Of course, coevolution is pervasive, but it may be possible that some functional processes coevolve even more than the baseline. We studied the overrepresentation of GO terms, KEGGs and reactome pathways. We found that several processes are overrepresented, some of which may represent species-specific heterochronies in cell differentiation or colonisation (eg by nerves and blood vessels), or size regulation (IGF1 pathway). Others are clearly associated with morphogenesis (cell and tissue migration). This is now mentioned in the manuscript and presented in Supplementary Figure 4

2. To ask whether coevolution participated in most or only in specific signalling pathways, or whether it concerned mostly regulators or their targets, we extracted signaling pathways and

their target genes from a previous study (O'Connell et al, 2012). We observed that the percentage of coevolution is high both in the signalling genes and in their targets, and that there is no significant difference between signalling pathways, even though FGF seems to coevolve a bit more. These results are now included in Figure 4A.

3. We have built the network formed by Fgf, Wnt, Bmp and their targets using the STRING database, and colored coevolving genes (in blue). The resulting plots are pasted below for BMP4, Wnt_mes (Wnt and its targets in the mesenchyme), Wnt_epi (Wnt and its targets in the epithelium). We did not include them as supplementary figures, because we did not observe a particular pattern for the coevolving genes, but we are ready to do so if that is a request from the reviewer.

Bmp4 pathway's targets (list from O'Connell et al. 2012)

Wnt pathway's targets in the mesenchyme (list from O'Connell et al. 2012)

Wnt pathway's targets in the epithelium (list from O'Connell et al. 2012)

- The observation that several developmental phenotypes of the lower molar have evolved in concert with the upper molar without driving major phenotypic changes is really important. Did the authors investigate whether some of the changes between mouse and hamster's lower molar transcriptome could, apparently paradoxically, have had an inhibitory effect on the generation of phenotypic changes? In other words, is it possible that the lack of major phenotypic changes in the lower molar is caused by the evolution of compensatory genetic networks?

R: We share this intuition with the Reviewer, that compensatory changes in the lower molar may have evolved and prevent phenotypic change. This grounds a new team project that has been ERC-funded and which we are just starting. Stay tuned !

- The analysis of co-evolution of transcriptomes during bat limb evolution would really benefit from a more extensive description and a more complete figure. I am fully aware that the article uses mostly the tooth as model and extends to the bat limb mostly to assess whether the principle identified during tooth evolution applies to other organs. Yet, this would make the case of the authors more compelling and more convincingly applicable to other organs (and it would be a pity to bury such an interesting analysis on the bat by using it so summarily)

R: We have added a new Figure (new Figure 6), in which we present the transcriptome dataset that was reanalysed, the models that were built with the percentage of coevolution, and three genes that are known to be associated with limb evolution in the literature. For these three genes (Shh, Fgf8 and Grem1), we present our transcriptomic models together with cartoons schematizing in situ hybridizations taken from several articles.

Thank you again!

POINT BY POINT ANSWER TO REVIEWERS' COMMENTS

Reviewer #1 (Remarks to the Author):

I have now read the revised manuscript and the rebuttal. The authors have done a thorough job in revising the work, and I thank them for addressing my lengthy comments. The figures are much improved, and the text is clearer. When I write clearer, it means that I would still wish the presentation to be less like a podium presentation in a large hall, and more like a meticulous exposition of scientific discovery. But I leave it to the authors to decide how much they want to tinker with the language. The first paragraph of the introduction remains particularly declaratory. But overall, this is a very valuable contribution.

ANSWER: we thank the reviewer for this frank comment and his tolerance of a style he doesn't appreciate. We insist that we very much appreciated his constructive feedback during the reviewing process, which benefited the manuscript.

Couple of oddities:

Page 12

“This may root in differences of basal expression levels or this may reflect co-evolution of the temporal dynamics of morphogenesis”

COMMENT: root?

ANSWER: This was changed as follow

This may be caused by differences of basal expression levels or this may reflect co-evolution of the temporal dynamics of morphogenesis.

End of the discussions:

“could retain pleiotropic mutations which impacted the development of both molars but poorly changed the lower molar”

COMMENT: Poorly? Maybe ‘but leave the morphology of lower molars relatively intact’ or perhaps ‘conserved’

ANSWER: This was changed as follow

We also propose concrete mechanisms for how selection could retain pleiotropic mutations which impacted the development of both molars but left the morphology of lower molars relatively intact.

Reviewer #1 (Remarks on code availability):

The github page looks all right, and the instructions are clear. I have not run the scripts.

Reviewer #2 (Remarks to the Author):

The authors addressed all my comments. Great work!

Thank you!